# Flatter Tokens are More Valuable for Speculative Draft Model Training

**Jiaming Fan**[1,2]* **Daming Cao**[3]* **Xiangzhong Luo**[1,2]† **Jiale Fu**[1,2] **Chonghan Liu**[4] **Xu Yang**[1,2]
[1]Key Laboratory of New Generation Artificial Intelligence Technology and Its Interdisciplinary Applications (Southeast University), Ministry of Education    [2]Southeast University
[3]Nanjing University of Information Science and Technology    [4]Qiyuan Tech
`jiaming.fan@seu.edu.cn`    `dmcao@nuist.edu.cn`
`xiangzhong.luo@seu.edu.cn`

## Abstract

Speculative Decoding (SD) is a key technique for accelerating Large Language Model (LLM) inference, but it typically requires training a draft model on a large dataset. We approach this problem from a data-centric perspective, finding that not all training samples contribute equally to the SD acceptance rate. Specifically, our theoretical analysis and empirical validation reveals that tokens inducing flatter predictive distributions from the target model are more valuable than those yielding sharply peaked distributions. Based on this insight, we propose `flatness`, a new metric to quantify this property, and develop the Sample-level-flatness-based Dataset Distillation (SFDD) approach, which filters the training data to retain only the most valuable samples. Experiments on the EAGLE framework demonstrate that SFDD can achieve over $2\times$ training speedup using only 50% of the data, while keeping the final model's inference speedup within 4% of the full-dataset baseline. This work introduces an effective, data-centric approach that substantially improves the training efficiency for Speculative Decoding. Our code is available at `https://github.com/fjm9933/Flatness`.

## 1 Introduction

Large language models (LLMs) have demonstrated remarkable success across a myriad of downstream tasks, such as generation, comprehension, and interaction (Achiam et al., 2023; Guo et al., 2025; Touvron et al., 2023). Despite the success, modern LLMs rely on autoregressive decoding, where each token must be generated in sequence based on all previous tokens. This inherently sequential process suffers from inferior parallelism, which leads to high latency and low throughput (Li et al., 2024a). A recent effort to tackle this dilemma is *speculative decoding* (SD) (Leviathan et al., 2023; Chen et al., 2023). SD leverages a small *draft* model to quickly generate multiple tokens, which are then verified in parallel by the larger *target* model. As a result, the target model can accept multiple tokens in a single forward pass without degrading the quality of generation, where higher acceptance rates can equivalently translate into better inference speedups.

The success of SD has subsequently inspired a plethora of SD variants, which can be broadly divided into *train-free* and *train-based* categories. Among them, train-free SD methods employ off-the-shelf lightweight LLMs as the drafter, which can offer simplicity and cost-effectiveness without additional training (Leviathan et al., 2023; Chen et al., 2023; Zhang et al., 2024a; Miao et al., 2024; Gong et al., 2024; Santilli et al., 2023). Nonetheless, these methods suffer from poor alignment between the draft and target models, which often results in low acceptance rates and frequent rollbacks. In contrast, train-based SD methods introduce an additional training to align the draft model with the target model, which can substantially improve acceptance rates compared to their train-free counterparts (Zhou et al., 2023; Li et al., 2024b; Cai et al., 2024; Elhoushi et al., 2024; Bachmann et al., 2025; Yi et al., 2024; Monea et al., 2023; Qin et al., 2024).

---

*Equal contribution
†Corresponding author

Despite the promising progress, current train-based SD methods still face critical limitations. In practice, these methods leverage vanilla knowledge distillation (KD) (Hinton et al., 2015) as the default strategy to align the draft model with the target model. However, a subtle yet fundamental discrepancy exists between the objectives of vanilla KD and SD: while vanilla KD minimizes the KL-divergence between the student (draft model) and teacher (target model) output distributions, SD focuses on maximizing the acceptance rate, which is theoretically linked to the $L_1$-norm between these two distributions, as proved in (Leviathan et al., 2023). Motivated by this theoretical insight, recent studies have investigated the direct use of the $L_1$-norm as an alternative training objective (Zhou et al., 2023). Nonetheless, empirical findings indicate that this approach is not consistently effective and can, in some cases, underperform even the standard KL-divergence-based distillation. While the precise reasons for these counterintuitive results remain unclear, this evidence strongly suggests that simply substituting the loss function is insufficient, and revisiting the question of which portions of the data actually provide the most meaningful training signal is warranted, rather than solely focusing on the choice of loss function.

We therefore revisit KD in the context of SD and introduce a simple theoretical model to reflect improvements in acceptance rate after a single KD step. We view one update of the draft distribution as a budget-limited move toward the teacher. This abstraction mirrors standard KD practice while allowing us to ask a concrete question: which target-side token distributions yield the largest acceptance-rate gains per unit of training? Our toy example analysis and empirical studies indicate a token-level insight: tokens with flatter target distributions (closer to uniform) deliver the most per-step reduction in the draft–target discrepancy that governs the acceptance rate, whereas sharply peaked tokens contribute little and saturate quickly. This reframes the importance of tokens for SD relative to classical KD: what matters is not only the choice of loss but also where the useful signal lies in the data. Significantly, this criterion depends only on the target model and can be computed offline, without warming up a draft model or tracking its changing predictions. Nevertheless, current training-based SD systems (e.g., the EAGLE series (Li et al., 2024b;c; Li et al.)) essentially train on all tokens, overlooking this heterogeneity and incurring avoidable overhead. These observations motivate filtering out low-value tokens to improve efficiency while preserving acceptance.

Guided by this principle, we introduce a practical `flatness` metric that scores each token by how close the target model's distribution is to uniform, instantiated with a simple cosine-based similarity. Empirical evaluations on real LLMs reveal that the `flatness` metric serves as a reliable headroom indicator of potential improvement: tokens with higher flatness (more uniform targets) undergo larger expected updates and yield substantial reductions in acceptance-related discrepancies. In contrast, tokens characterized by low flatness (i.e., sharply peaked distributions) contribute minimally. This clear distinction consistently emerges under a target-sorted perspective. We then aggregate token-level scores to the sample level, enabling an effective data-selection approach. We term this approach *Sample-level-flatness-based Dataset Distillation* (SFDD), which yields a simple pipeline: (i) run a single offline pass of the target model to compute sample-level-flatness, (ii) rank and retain the high-value samples, and (iii) train the draft model on the filtered data. Plugged into EAGLE-2 (Li et al., 2024c), our selection preserves speedup while substantially reducing training time, and it outperforms common data selection metrics, such as entropy (Li et al., 2021), top-1 probability (Hendrycks & Gimpel, 2016), the margin between the top two probabilities (Kremer et al., 2014; Bahri et al., 2022), Energy Score (Liu et al., 2020), and perplexity (PPL) (Chen & Goodman, 1999; Meister & Cotterell, 2021). Finally, we summarize our main contributions as follows:

- **Revisiting KD for SD with an acceptance-centric lens.** We analyze a single KD step through a budget-limited update toward the teacher and show that tokens with flatter target distributions are disproportionately valuable for improving acceptance, whereas highly peaked tokens offer diminishing returns. Crucially, the resulting importance criterion depends only on the target model, allowing for offline scoring without a warmed-up student.

- **A simple, empirically strong importance metric.** We propose `flatness` as a practical proxy for token and sample importance in SD training, and demonstrate that it outperforms previous heuristics for identifying high-value training data on sample selection.

- **An efficient data-selection method for train-based SD.** Our SFDD method is effective across various data retention ratios. For example, at 50% retain ratio, we achieve over $2\times$ training speedup using only half the data, while also significantly outperforming other selection metrics and keeping the final model's inference speedup within 4% of the full-dataset baseline.

## 2 RELATED WORK

**Speculative decoding**. Speculative Decoding (SD) accelerates autoregressive generation via a "draft-and-verify" paradigm, with approaches broadly categorized as training-free or train-based methods. (1) Training-Free SD requires no new parameters, instead modifying inference via methods like rejection sampling (Leviathan et al., 2023; Chen et al., 2023), reusing target model parts (Zhang et al., 2024a), or parallel candidate verification (Miao et al., 2024; Gong et al., 2024; Santilli et al., 2023). While easily deployable, these reliance on heuristics often results in limited acceptance rates. (2) Train-Based SD fine-tunes a draft model for better alignment, either through distillation (Zhou et al., 2023; Goel et al., 2024) or by augmenting the target model with trainable components, including lightweight prediction heads (Li et al., 2024b;c; Li et al.; Cai et al., 2024), trainable early exits (Elhoushi et al., 2024), or auxiliary modules for multi-token prediction or lenient verification (Bachmann et al., 2025; Yi et al., 2024; Monea et al., 2023; Qin et al., 2024). These methods provide significantly higher and more stable speedups. In addition, some recent works (Zhou et al., 2023; Goel et al., 2024) leverage the theoretical objective (e.g., total variation distance) to improve alignment; however, they focus on loss function optimization to improve alignment. In contrast, our work targets efficient draft model training from the perspective of dataset distillation, remaining orthogonal to these train-based methods that introduce non-trivial training and deployment overhead.

**Data importance measurement methods**. Following trends in selective learning and aligned controllable generation (Zhu et al., 2024a;b; 2025; Zhao et al., 2025a;b), we focus on data importance from two perspectives: (1) Distributional Uncertainty: This approach gauges importance via the model's uncertainty, based on the intuition that the most informative samples are those the model is unsure about. This is often quantified by metrics such as the Shannon entropy of the output distribution (Li et al., 2021; Wang et al., 2025), the Energy Score derived from logits (Liu et al., 2020), or the sample difficulty measured by Perplexity (PPL) (Chen & Goodman, 1999; Meister & Cotterell, 2021). (2) Category Probability: This second approach derives importance from salient category probabilities. It includes using the Top-1 Probability or logit as a direct measure of model confidence (Hendrycks & Gimpel, 2016; Zhou et al., 2025), the margin between the top two probabilities $(p_{(1)} - p_{(2)})$ to gauge ambiguity (Kremer et al., 2014; Bahri et al., 2022), or the ground-truth token's probability to up-weight difficult examples (Lin et al., 2017; Kim et al., 2023). However, these heuristics are generally designed for standard training objectives like improving model accuracy or distribution fidelity. In contrast, our work is the first to systematically investigate data importance from the unique perspective of SD, where the central focus is the token acceptance rate.

## 3 PRELIMINARIES

### 3.1 SPECULATIVE DECODING

As shown in (Leviathan et al., 2023; Chen et al., 2023), speculative decoding (SD) utilizes a small, fast draft model and a large, powerful target model. The process begins with the draft model autoregressively generating a sequence of $\gamma$ candidate tokens, which are then verified in parallel by the target model. Formally, given a context of previously generated tokens $h$, we denote the candidate probability distribution from the draft model as $q(\cdot|h)$ and the reference distribution from the target model as $p(\cdot|h)$. Each candidate token $y$ is validated via rejection sampling and accepted with a probability of $\beta(y) = \min\left(1, \frac{p(y|h)}{q(y|h)}\right)$. If a candidate is rejected, the generation process is rolled back to the last accepted token. A new token is then drawn from the residual distribution $r(\cdot \mid h) \propto \left[p(\cdot \mid h) - q(\cdot \mid h)\right]_+$ (where $[x]_+ = \max\{x, 0\}$) to ensure that the final output is equivalent to sampling directly from the target distribution $p(\cdot|h)$.

**Acceptance rate.** In SD, the acceptance rate $\alpha(h)$ is a key performance metric, defined as the ratio of candidate tokens proposed by the draft model that are verified and accepted by the target model. In practice, higher acceptance rates can equivalently translate into better inference speedups. Prior work (Leviathan et al., 2023) has shown that the acceptance rate is linearly decreasing in the $L_1$-norm between the target model's distribution $p$ and the draft model's distribution $q$:

$$\alpha(h) = \mathbb{E}_{y \sim q(\cdot|h)}[\beta(y)] = \sum_y \min\{p(y \mid h), q(y \mid h)\} = 1 - \tfrac{1}{2}\big\|p(\cdot \mid h) - q(\cdot \mid h)\big\|_1. \quad (1)$$

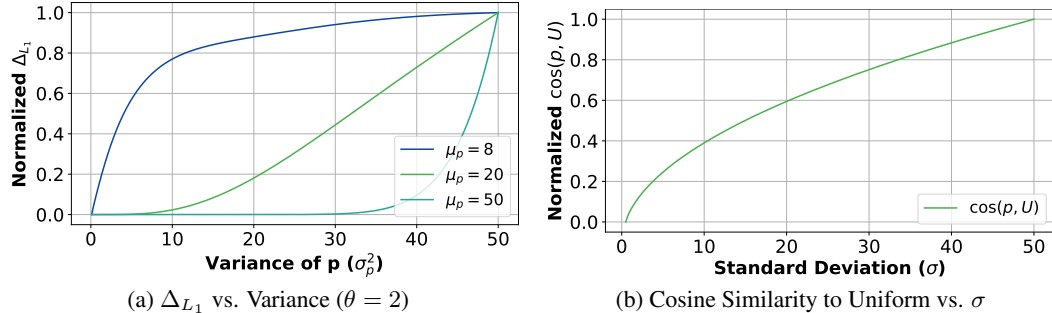

(a) $\Delta_{L_1}$ vs. Variance ($\theta = 2$)   (b) Cosine Similarity to Uniform vs. $\sigma$

Figure 1: **Simulation of importance metrics.** We use Gaussian distributions in our simulation because they are analytically tractable and effectively capture key distributional properties like variance. For visualization clarity, all plotted quantities are min–max normalized to $[0, 1]$. (a) Across small, medium, and large separations between the means of $p$ and $q$, $\Delta_{L_1}$ consistently increases with the target variance ($\sigma_p^2$). (b) Cosine similarity to uniform increases monotonically with the standard deviation ($\sigma$), validating its use as a practical proxy that directly tracks changes in variance.

Thus, a per-token improvement in acceptance equals a per-token reduction in the $L_1$-norm distance. In light of this, maximizing the token-level acceptance rate is strictly equivalent to minimizing the $L_1$-norm distance between the output distributions of the target model and draft model.

## 3.2 THEORETICAL ANALYSIS OF KD IN SD

In this section, we investigate knowledge distillation (KD) in the context of SD. We start from the optimization objective of SD and then present both empirical analysis and experimental validation.

**Training objective.** An intuitive approach to optimizing the acceptance rate is to directly use the $L_1$-norm as a training loss. However, this can yield suboptimal results on certain tasks compared to the conventional KL-divergence objective (Zhou et al., 2023). This further leads us to investigate how the characteristics of the target model's token distribution $p$ and the draft model's token distribution $q$ influence the $L_1$-norm. Analyzing the draft model's distribution $q$ is challenging, as it is a moving target during training. The target distribution $p$, however, remains fixed within a given context. This stability allows us to pursue an empirical goal: identifying the properties of an optimal target distribution $p$ that are beneficial regardless of the specific state of $q$.

At the same time, a small $L_1$-norm does not necessarily indicate a valuable training opportunity. For instance, if $q$ is already very close to $p$, the $L_1$-norm will be minimal, but training on this token will also yield a negligible contribution to the model's improvement. This insight suggests that the true measure of a token's value is not the static $L_1$-norm itself, but the potential training contribution. We therefore quantify this contribution as the reduction in the $L_1$-norm achieved in a single training step, denoted as $\Delta_{L_1}$. Formally, given an initial draft distribution $q$ and an updated distribution $r^*$ after one step, this contribution is defined as:

$$\Delta_{L_1} = \|p - q\|_1 - \|p - r^*\|_1. \tag{2}$$

Our primary objective is to explore characteristics of the target token distribution $p$ that maximize $\Delta L_1$ for a given draft token distribution $q$. To study this, we introduce $r^*$ as a theoretical proxy for the draft distribution after a *single, idealized* update. This theoretical toy model serves as a simplified analytical model to illustrate how a budget-constrained training step might ideally shift the draft distribution toward the target. Notably, $r^*$ is not employed as a practical training target; instead, it functions solely as an analytical tool to understand incremental improvements.

Modeling a single training step inevitably involves simplifications, and various formalisms could potentially serve this purpose. We select an approach closely aligned with standard KD practices, yet analytically tractable: our *objective* adheres to KD by minimizing $D_{\mathrm{KL}}(p\|r)$, while we simultaneously impose a small-step *budget* constraint to reflect that an update must remain close to the original draft distribution $q$. Crucially, our insights do not depend strongly on the specific choice of budget measurement; alternative measures such as $L_p$ norm, Jensen–Shannon divergence, or other suitable metrics would yield similar conclusions. We adopt KL divergence $D_{\mathrm{KL}}(r\|q)$ here, as this choice facilitates a concise and explicit analytical form for $r^*$ under the subsequent Gaussian setting,

thus ensuring that our downstream analysis remains transparent and interpretable:

$$r^* = \arg\min_r D_{\mathrm{KL}}(p\|r) \quad \text{s.t.} \quad D_{\mathrm{KL}}(r\|q) \leq \theta, \tag{3}$$

where $\theta \geq 0$ plays the role of a step-size budget (capturing, e.g., learning-rate and optimizer effects) that limits how far $r$ can deviate from $q$ in one update.

**Solution for parametric Gaussian distributions**. Analyzing the optimal distribution $r^*$ in its general non-parametric form is analytically challenging. To obtain analytical insights, we first restrict the above optimization problem to the parametric family of Gaussian distributions. Specifically, we use $p = \mathcal{N}(\mu_p, \sigma_p^2)$ to denote the target token distribution and $q = \mathcal{N}(\mu_q, \sigma_q^2)$ to denote the draft token distribution. Using the Karush-Kuhn-Tucker (KKT) conditions (see Appendix A for the detailed derivation), we solve for the optimal distribution $r^* = \mathcal{N}(\mu_r^*, \sigma_r^{2*})$, whose parameters are:

$$\mu_r^* = (1 - \tau^*)\mu_p + \tau^*\mu_q, \tag{4}$$

$$\sigma_r^{2*} = (1 - \tau^*)\sigma_p^2 + \tau^*\sigma_q^2 + \tau^{*2}(1 - \tau^*)(\mu_p - \mu_q)^2. \tag{5}$$

The optimal distribution $r^*$ is found by minimizing $D_{\mathrm{KL}}(p\|r)$ under the constraint that $D_{\mathrm{KL}}(r\|q) \leq \theta$. It lies on a path between $p$ and $q$, and its position on this path is determined by a single parameter $\tau^* \in [0, 1]$. This parameter quantifies the extent of the update in a single training step: $\tau^* = 0$ corresponds to a full update where $r^*$ becomes $p$, while $\tau^* = 1$ means no update has occurred ($r^*$ remains $q$). The specific value of $\tau^*$ is uniquely determined by the training budget $\theta$.

**Simulation results.** Although the above solution provides an analytical form for the optimal parameters $(\mu_r^*, \sigma_r^{2*})$, the path parameter $\tau^*$ itself lacks a closed-form solution and must be determined numerically. Therefore, to investigate the properties of this solution, we establish a numerical simulation, where the draft distribution $q$ is fixed as the standard normal distribution ($\mu_q = 0, \sigma_q^2 = 1$) and the training budget is a fixed $\theta$. For various target token distributions $p$ (defined by sweeping their mean $\mu_p$ and variance $\sigma_p^2$), we first solve for the optimal path parameter $\tau^*$, based on which we can derive the parameters of $r^*$. With the updated distribution $r^*$ now fully defined, we can substitute it back into the expression for $\Delta_{L_1}$ to analyze which properties of $p$ lead to the largest training benefit. Our simulation results are illustrated in Figure 1a, which shows how the $L_1$-norm distance reduction $\Delta_{L_1}$ varies with respect to the target distribution, $\sigma_p^2$. Within our tested range of variances, we observe a clear trend: for a given mean $\mu_p$, the reduction $\Delta_{L_1}$ tends to increase as $\sigma_p^2$ grows. This suggests that tokens whose target distributions have higher variance are likely to yield the greatest reduction in the $L_1$-norm during training. The empirical results from our simulation reveal a clear relationship: tokens with larger variance $\sigma_p^2$ are more valuable to the training process.

This finding can also be explained by our formula. As shown in Equation 5, a larger target variance $\sigma_p^2$ directly increases the variance of the updated distribution $r^*$, ensuring a flatter target yields a flatter updated distribution. This shape alignment is crucial for maximizing the training contribution, $\Delta_{L_1}$ (Equation 2). The $L_1$-norm is highly sensitive to the misalignment of sharp probability peaks; since flat distributions lack such peaks, pointwise differences between them remain small, resulting in a smaller distance $\|p - r^*\|_1$. Assuming a constant initial distance $\|p - q\|_1$ (i.e., a fixed starting acceptance rate), minimizing $\|p - r^*\|_1$ maximizes the training contribution $\Delta_{L_1}$ (Equation 2).

**Key insights.** This motivates our key theoretical insight: not all tokens are equally important; those with flatter target distributions are more valuable for training. And we can use the variance of the Gaussian distribution as a measure of token importance in SD.

**Discrete perspective.** However, in the discrete, token-level probability distributions produced by practical LLMs, we cannot directly compute this continuous variance. To bridge this gap between continuous theory and discrete distributions, we require a useful metric that can serve as a proxy for variance. We propose using the **cosine similarity with the uniform distribution.** This choice is theoretically grounded; as detailed in Appendix B, it can be shown that this metric is positively correlated with the variance of the corresponding Gaussian distribution in the continuous limit. Our simulations, presented in Figure 1b, further validate this crucial relationship, demonstrating that the cosine similarity indeed increases monotonically with the Gaussian standard deviation. This positive correlation is the crucial link between our continuous theory and discrete application. It validates that cosine similarity to a uniform distribution can serve as a practical and computable proxy for quantifying a token's training importance in the discrete setting.

# 4 THE PROPOSED APPROACH

## 4.1 EMPIRICAL VALIDATION OF THE THEORETICAL ANALYSIS

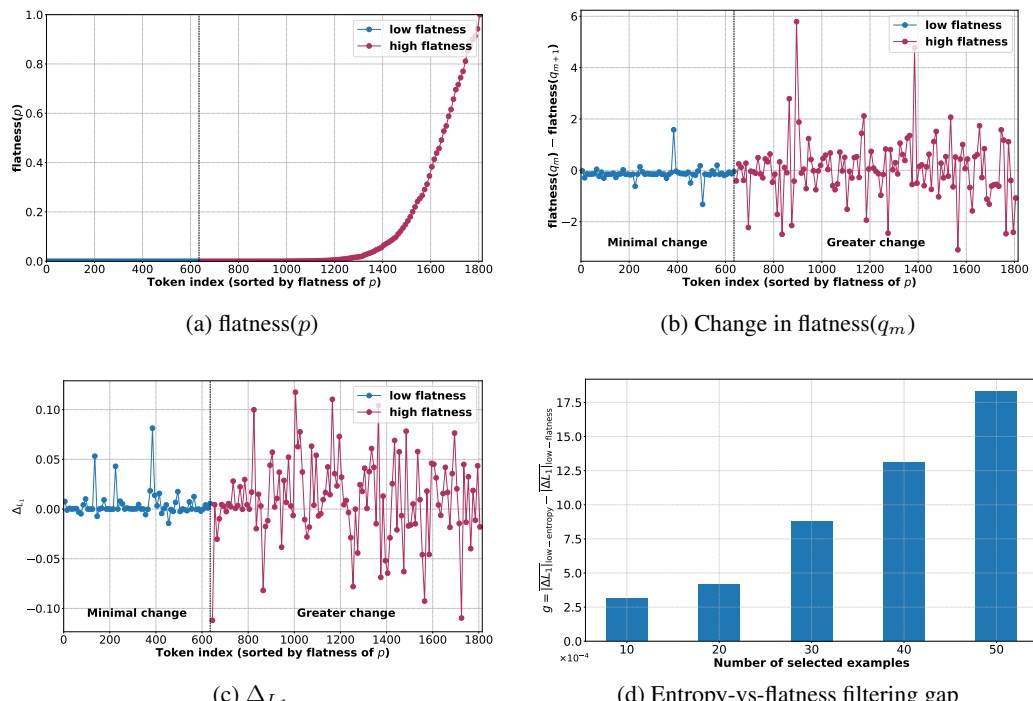

(a) flatness($p$)

(b) Change in flatness($q_m$)

(c) $\Delta_{L_1}$

(d) Entropy-vs-flatness filtering gap

Figure 2: **Target-sorted flatness view.** Tokens sorted by the target's statistic (low→high flatness). (a) flatness values; (b) one-epoch change in flatness; (c) the one-epoch reduction in the $L_1$ discrepancy, $\Delta_{L_1}$. Curve coloring distinguishes token groups by target flatness: the **blue** segment represents tokens with low flatness, while the **red** segment represents those with high flatness. Panels (b,c) additionally annotate **Minimal change** (left; indicating that the vast majority of points in this segment exhibit small changes) and **Greater change** (right; indicating that the vast majority of points in this segment exhibit larger changes). (d) Entropy-vs-flatness filtering gap: for each metric, we rank tokens by that metric, take the bottom 35% tokens, and compute their one-epoch average $|\Delta_{L_1}|$; bars plot the difference between the entropy-based and flatness-based bottom-35% averages under different numbers of selected examples. Flatness is as defined in Equation 6.

**Flatness definition.** In the previous section, our one-step KD analysis reveals that tokens with flatter target distributions should offer greater acceptance-linked headroom; however, those results rest on idealized assumptions. Therefore, an empirical validation is essential to bridge the gap between our theory and real-world application. So now we test this insight on real LLMs. We define `flatness` of a token $t$ as the cosine similarity between its distribution and the uniform distribution:

$$flatness(t) := \cos(p_t, U) = \frac{p_t \cdot U}{\|p_t\|_2 \|U\|_2}, \tag{6}$$

where $p_t$ is the token's distribution, $U$ is the uniform distribution over the vocabulary of size $V$, and $\| \cdot \|_2$ denotes the Euclidean ($L_2$) norm.

**Empirical validation via training dynamics.** From the definition, a higher flatness denotes a more uniform distribution, and a lower flatness indicates a more concentrated distribution. When we refer to the target flatness, we use the target distribution $p$ as $p_t$ in the equation. To validate whether target flatness can effectively serve as a metric for a token's training potential, we first need a metric to quantify its actual contribution to the learning process. Throughout our analysis, we define this contribution as the epoch-to-epoch reduction in the $L_1$ discrepancy, i.e.,

$$\Delta_{L_1} = \|p - q_m\|_1 - \|p - q_{m+1}\|_1, \tag{7}$$

where $m$ and $m+1$ denote consecutive training epochs. We follow the EAGLE-2 framework (Li et al., 2024c) with LLaMA3-8B-Instruct (Grattafiori et al., 2024) as target model, trained on filtered ShareGPT dataset (sha, 2023). And we randomly select 10 samples for detailed inspection.

To relate the evaluated metric to training progression, we first sort tokens by the target flatness value in ascending order (low to high). The resulting curves are then smoothed using a 10-point moving average to obtain more stable values, thereby enhancing readability. As shown in Figure 2, we plot three key metrics. The x-axis for all subplots represents tokens sorted in ascending order of the target flatness, $flatness(p)$. To analyze the draft model's behavior, we apply this flatness metric to its distribution $q$, which we called draft flatness. The plots then show: (a) the target flatness value itself; (b) the one-epoch change in the draft model's flatness, $flatness(q_m)$; and (c) the corresponding one-epoch reduction in the $L_1$ discrepancy, $\Delta_{L_1}$.

Collectively, our empirical results substantiate the following observations, sorting tokens according to the target flatness from low to high, as illustrated in the target-sorted view in Figure 2:

**(i)** In the *low target flatness* region (blue segment in Panel (a)), the one-epoch change in draft flatness is small (Panel (b), left; annotated "Minimal change"), and the acceptance-linked discrepancy likewise shows slight movement (Panel (c), left). Thus, tokens with low target flatness exhibit *limited* one-epoch movement in both draft statistics and $\Delta_{L_1}$.

**(ii)** In the *high target flatness* region (red segment in Panel (a)), draft flatness varies more over one epoch (Panel (b), right; annotated "Greater change"), and the magnitude of the corresponding change in $\Delta_{L_1}$ is also larger (Panel (c), right). Hence, tokens with high target flatness are precisely where we observe pronounced acceptance-linked movement during training.

These findings affirm that flatness effectively signals available headroom: tokens with low target flatness contribute minimally to training improvements, whereas tokens with high target flatness exhibit learnable dynamics and significant changes in $\Delta_{L_1}$. We thus adopt target flatness ($flatness(p)$) as the principal ranking criterion for data selection.

At first glance, target flatness-based selection might seem counterintuitive or risky: interpreting low target flatness as indicative of high target certainty (a strong label signal) could imply inadvertently excluding valuable tokens. In practice, however, such low target flatness tokens either (i) already closely align or will rapidly align with the target distribution, rendering subsequent updates negligible in terms of reducing $\Delta_{L_1}$ in later training, or (ii) remain confidently misaligned, thus providing minimal per-step gradient information and potentially contributing negatively when averaged across multiple tokens. Consequently, prioritizing tokens with higher target flatness focuses computational resources precisely where meaningful, acceptance-linked improvements can be realized.

**Comparison with other metrics.** We further compare target flatness with other commonly used distribution-dispersion metrics, such as target entropy. The results show a similar trend, details are provided in Appendix F.2.

More importantly, we find that flatness provides more effective filtering than entropy. We randomly sample $N \in \{10, 20, 30, 40, 50\}$ training examples. For each metric (entropy or flatness), we rank all tokens by that metric and take the bottom 35% as low-score tokens. On the low-entropy and low-flatness tokens of each metric, we compute the average $\overline{|\Delta_{L_1}|}$ between consecutive training epochs, denoted as $\overline{|\Delta_{L_1}|}_{\text{low-entropy}}$ and $\overline{|\Delta_{L_1}|}_{\text{low-flatness}}$, for their remaining impact on SD in late training. We then report the gap between flatness and entropy as $g = \overline{|\Delta_{L_1}|}_{\text{low-entropy}} - \overline{|\Delta_{L_1}|}_{\text{low-flatness}}$.

As shown in Figure 2d, we observe $g > 0$ consistently holds. Furthermore, the gap increases as $N$ grows. This indicates that, under the same retain ratio, flatness-based filtering removes more already-saturated tokens (with smaller $|\Delta_{L_1}|$). It gives a quantitative explanation of why flatness is a better metric than entropy in our data selection experiments: flatness is more effective at filtering out low-quality tokens that offer minimal training value, leading to higher training efficiency.

## 4.2 FROM TOKEN-LEVEL INSIGHT TO SAMPLE-LEVEL DATA SELECTION

The successful validation of this token-level insight sheds light on a significant inefficiency in current training-based SD methods. Prominent approaches, such as the EAGLE series, rely on full-data training over large datasets. They treat all data samples as equally important, thereby expending con-

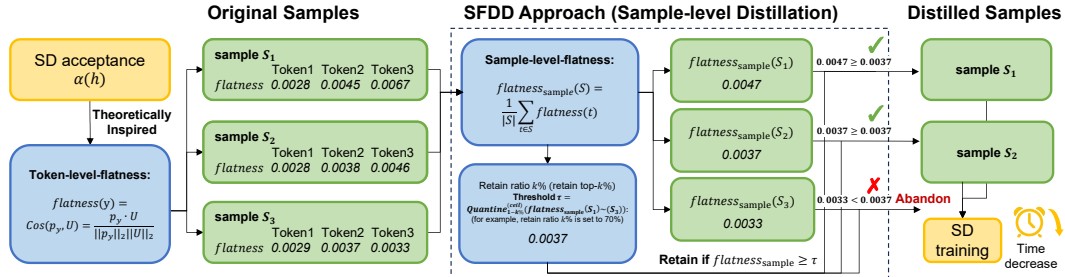

Figure 3: **The SFDD workflow:** This approach calculates $flatness_{\text{sample}}$ by averaging token flatness within each sample, and then uses quantile-derived threshold to select the top-$k\%$ and filter the dataset for training. The figure illustrates this with a concrete example using 70% retain ratio.

siderable computational resources on samples predominantly composed of low-value, concentrated tokens, while these tokens contribute negligibly to training. This motivates us to advocate for a more efficient paradigm: extending our validated token-level insight to a practical, sample-level data selection strategy. The goal is to filter out entire samples that are unlikely to contribute significantly to the training process, thereby accelerating training without heavily sacrificing performance.

To this end, we introduce the sample-level-flatness to quantify a sample's overall training value. The sample-level-flatness for a sample $S$ is defined as the average of the flatness of its constituent tokens:

$$flatness_{\text{sample}}(S) = \frac{1}{|S|} \sum_{t \in S} flatness(t), \tag{8}$$

where the token-level $flatness(t)$ is calculated using the target distribution $p$. A higher $flatness_{\text{sample}}$ signifies that the sample, as a whole, offers greater potential training value.

### 4.3 SAMPLE-LEVEL-FLATNESS-BASED DATASET DISTILLATION

Building on our validated sample-level flatness metric, we introduce a simple yet effective approach for dataset distillation. This approach curates a smaller, more efficient dataset for SD training by retaining only samples with high flatness. The overall workflow, which we term *Sample-level-flatness-based Dataset Distillation* (SFDD), is illustrated in Figure 3. Given a retain ratio of $k\%$, the procedure is to first compute sample-level flatness for each sample by averaging the flatness of its constituent tokens. A threshold $\tau$ is then set as the ceiling of the $(1-k)\%$ quantile of these scores. The distilled dataset is formed by retaining all samples with $flatness_{\text{sample}} \geq \tau$.

## 5 EXPERIMENTS

In the previous sections, our theoretical and empirical findings have shown that a token's importance correlates with the $flatness$ of its target distribution. In this section, we extensively evaluate our approach, Sample-level-flatness-based Dataset Distillation (SFDD), against various baselines across different datasets. We aim to: (1) demonstrate the superiority of SFDD by benchmarking it against various common importance metrics at a fixed data retain ratio; (2) compare SFDD against a random filtering baseline across different data retain ratios to demonstrate the effectiveness of our method; and (3) quantify the training efficiency gains provided by our data selection approach.

### 5.1 EXPERIMENTAL SETUP

**Models and baselines.** Our experiments use the EAGLE-2 training pipeline (Li et al., 2024c) with LLaMA3-8B-Instruct (Grattafiori et al., 2024) as the target model. Our approach is compared against two main baselines: training on the full dataset ("No Filter") and a naive "Random Filtering" strategy. In our main results, we also benchmark against several other common token-importance metrics, including entropy (Li et al., 2021), Top-1 Probability (Hendrycks & Gimpel, 2016), the margin between the top two probabilities ($p_{(1)} - p_{(2)}$) (Kremer et al., 2014; Bahri et al., 2022), Energy Score (Liu et al., 2020), and perplexity (PPL) (Chen & Goodman, 1999; Meister & Cotterell, 2021). For these metrics, we select samples with high entropy, low top-1 probability, small margin,

larger (less-confident) Energy Score, or higher PPL, as these criteria help identify samples that are valuable for training rather than those the model has already converged on.

**Dataset and tasks.** We use the ShareGPT dataset (sha, 2023) for training. Evaluation is performed on five diverse downstream tasks: GSM8K (Cobbe et al., 2021), Alpaca (Taori et al., 2023), MT-Bench (MTB) (Zheng et al., 2023), CNN/DM (See et al., 2017), and Natural Questions (NQ) (Kwiatkowski et al., 2019). All experiments use NVIDIA H800 GPUs, a decoding temperature of 1.0, and a draft generating step of $\gamma = 5$; see Appendix C for temperature 0 results.

**Evaluation metrics.** We use three primary metrics: (1) **Speedup**: The wall-clock time of standard autoregressive decoding divided by that of speculative decoding. Higher is better. (2) **Average acceptance length** ($l$): The average number of draft tokens accepted per verification cycle. The average acceptance rate is exactly $l/\gamma$, but since the rate is often very small, we instead report $l$, which makes variations more visually discernible. Higher is better. (3) **Training time**: The total wall-clock time (in seconds) for training, used to measure efficiency. All reported times include data-selection overhead. More details can be found in Appendix D.

## 5.2 MAIN RESULTS

We fix the data retain ratio at 50% for our main comparison. This ratio is chosen because our preliminary experiments show that the random filtering baseline performs near its peak at this level, ensuring a fair comparison against a strong baseline. We compare our method against several common metrics that, similar to our approach, measure data importance, reporting both inference speedup and average generation length. As shown in Table 1, our SFDD method consistently achieves higher speedup and average generation length than all other metrics across every downstream task. Notably, SFDD achieves an average speedup of $2.41\times$, which is significantly higher than the next best method (Top-1 Probability at $2.23\times$). Furthermore, with only 50% of the data, our method exhibits the smallest performance gap compared to the full-dataset baseline, achieving an average speedup that is **within 4% of the "No Filter" speedup ($2.41\times$ vs. $2.49\times$).**

Table 1: Comprehensive comparison of various metrics for data importance at a 50% retain ratio.

| Method | GSM8K | | Alpaca | | MTB | | CNN/DM | | NQ | | Average | |
|---|---|---|---|---|---|---|---|---|---|---|---|---|
| | Speedup | $l$ | Speedup | $l$ | Speedup | $l$ | Speedup | $l$ | Speedup | $l$ | Speedup | $l$ |
| No Filter | $2.71\times$ | 3.28 | $2.71\times$ | 2.89 | $2.53\times$ | 2.77 | $2.30\times$ | 2.58 | $2.19\times$ | 2.37 | $2.49\times$ | 2.78 |
| Random | $2.43\times$ | 2.85 | $2.37\times$ | 2.59 | $2.26\times$ | 2.48 | $1.99\times$ | 2.31 | $1.93\times$ | 2.06 | $2.20\times$ | 2.46 |
| Entropy | $2.43\times$ | 2.85 | $2.43\times$ | 2.64 | $2.20\times$ | 2.51 | $1.95\times$ | 2.32 | $1.98\times$ | 2.12 | $2.20\times$ | 2.49 |
| Top-1 Probability | $2.49\times$ | 2.84 | $2.44\times$ | 2.66 | $2.26\times$ | 2.53 | $1.99\times$ | 2.32 | $1.98\times$ | 2.12 | $2.23\times$ | 2.49 |
| Margin | $2.45\times$ | 2.85 | $2.35\times$ | 2.48 | $2.19\times$ | 2.42 | $1.92\times$ | 2.27 | $1.85\times$ | 1.99 | $2.15\times$ | 2.40 |
| Energy Score | $2.49\times$ | 2.87 | $2.44\times$ | 2.64 | $2.19\times$ | 2.50 | $1.99\times$ | 2.33 | $1.91\times$ | 2.12 | $2.21\times$ | 2.49 |
| PPL | $2.36\times$ | 2.79 | $2.45\times$ | 2.65 | $2.21\times$ | 2.50 | $2.01\times$ | 2.33 | $1.95\times$ | 2.13 | $2.20\times$ | 2.48 |
| **SFDD (Ours)** | $\mathbf{2.69\times}$ | **2.95** | $\mathbf{2.66\times}$ | **2.71** | $\mathbf{2.44\times}$ | **2.60** | $\mathbf{2.14\times}$ | **2.38** | $\mathbf{2.14\times}$ | **2.17** | $\mathbf{2.41\times}$ | **2.56** |

## 5.3 ABLATION STUDY

We conduct an ablation study, with results in Table 2, to ablate two key factors. The setup allows us to simultaneously ablate the contribution of our selection metric (by contrasting SFDD with Random Filtering and Top-1 Probability, which is the second-best metric in Table 1), ablate the impact of the retain ratio and control for the specificity of the data chosen by random filtering (by observing the trend across different retain ratios), thereby confirming that our method's advantage is robust and not coincidental. The results in Table 2 demonstrate two key points. First, SFDD surpasses both the random baseline and Top-1 Probability by a large margin across all retain ratios, confirming the effectiveness of our flatness-based scoring metric. Second, a significant speedup gap between SFDD and the baselines persists even at low retain ratios, highlighting the effectiveness of our method. Impressively, with 70% of the data, our method's speedup is nearly identical to the "No Filter" baseline, and on certain datasets like Alpaca, it even surpasses the full-dataset speedup ($2.77\times$ vs. $2.71\times$), suggesting that filtering can sometimes remove noisy or redundant data.

To investigate the robustness of our method under highly resource-constrained scenarios, we extend our evaluation to include extreme retain ratios of 5%, 10%, and 20%. We compare SFDD against Random. As shown in Table 3, we observe that while both methods experience a performance drop

Table 2: Ablation study on different retain ratios, comparing SFDD against different baselines.

| Retain Ratio | Method | GSM8K | | Alpaca | | MTB | | CNN/DM | | NQ | | Average | |
|---|---|---|---|---|---|---|---|---|---|---|---|---|---|
| | | Speedup | $l$ | Speedup | $l$ | Speedup | $l$ | Speedup | $l$ | Speedup | $l$ | Speedup | $l$ |
| 100% | No Filter | 2.71× | 3.28 | 2.71× | 2.89 | 2.53× | 2.77 | 2.30× | 2.58 | 2.19× | 2.37 | 2.49× | 2.78 |
| 70% | Random | 2.43× | 2.84 | 2.41× | 2.55 | 2.24× | 2.46 | 1.98× | 2.30 | 1.91× | 2.06 | 2.19× | 2.44 |
| | Top-1 Probability | 2.62× | 2.89 | 2.61× | 2.70 | 2.34× | 2.50 | 2.07× | 2.37 | 2.09× | 2.14 | 2.35× | 2.52 |
| | **SFDD (Ours)** | **2.71×** | **2.95** | **2.77×** | **2.77** | **2.41×** | **2.58** | **2.19×** | **2.40** | **2.14×** | **2.19** | **2.44×** | **2.58** |
| 60% | Random | 2.42× | 2.89 | 2.38× | 2.59 | 2.22× | 2.49 | 2.02× | 2.32 | 1.95× | 2.06 | 2.20× | 2.47 |
| | Top-1 Probability | 2.54× | 2.92 | 2.55× | 2.67 | 2.35× | 2.51 | 2.07× | 2.37 | 2.09× | 2.14 | 2.32× | 2.52 |
| | **SFDD (Ours)** | **2.55×** | **2.95** | **2.71×** | **2.72** | **2.40×** | **2.57** | **2.15×** | **2.40** | **2.13×** | **2.15** | **2.39×** | **2.56** |
| 50% | Random | 2.43× | 2.85 | 2.37× | 2.59 | 2.26× | 2.48 | 1.99× | 2.31 | 1.93× | 2.06 | 2.20× | 2.46 |
| | Top-1 Probability | 2.49× | 2.84 | 2.44× | 2.66 | 2.26× | 2.53 | 1.99× | 2.32 | 1.98× | 2.12 | 2.23× | 2.49 |
| | **SFDD (Ours)** | **2.69×** | **2.95** | **2.66×** | **2.71** | **2.44×** | **2.60** | **2.14×** | **2.38** | **2.14×** | **2.17** | **2.41×** | **2.56** |
| 40% | Random | 2.47× | 2.85 | 2.39× | 2.57 | 2.24× | 2.46 | 1.99× | 2.31 | 1.87× | 2.06 | 2.19× | 2.45 |
| | Top-1 Probability | 2.41× | 2.84 | 2.44× | 2.63 | 2.26× | 2.50 | 1.99× | 2.32 | 1.93× | 2.08 | 2.20× | 2.47 |
| | **SFDD (Ours)** | **2.66×** | **2.94** | **2.63×** | **2.73** | **2.40×** | **2.56** | **2.13×** | **2.36** | **2.12×** | **2.14** | **2.39×** | **2.55** |
| 30% | Random | 2.31× | 2.79 | 2.33× | 2.55 | 2.19× | 2.41 | 1.99× | 2.23 | 1.88× | 2.01 | 2.14× | 2.40 |
| | Top-1 Probability | 2.40× | 2.83 | 2.40× | 2.62 | 2.23× | 2.43 | 1.95× | 2.27 | 1.92× | 2.07 | 2.18× | 2.44 |
| | **SFDD (Ours)** | **2.51×** | **2.88** | **2.60×** | **2.62** | **2.37×** | **2.50** | **2.17×** | **2.33** | **2.01×** | **2.10** | **2.33×** | **2.49** |

at these very low retain ratios, SFDD consistently outperforms Random filtering across all datasets in terms of both inference speedup and average acceptance length ($\ell$). This indicates that flatness remains a robust and effective indicator of token importance under extreme data reduction regimes.

Table 3: Ablation study at extreme retain ratios, comparing SFDD against Random filtering.

| Retain | Method | GSM8K | | Alpaca | | MTB | | CNN/DM | | NQ | | Average | |
|---|---|---|---|---|---|---|---|---|---|---|---|---|---|
| | | Speedup | $\ell$ | Speedup | $\ell$ | Speedup | $\ell$ | Speedup | $\ell$ | Speedup | $\ell$ | Speedup | $\ell$ |
| 5% | Random | 1.75× | 1.96 | 2.00× | 1.92 | 1.60× | 1.73 | 1.48× | 1.45 | 1.57× | 1.49 | 1.68× | 1.71 |
| | **SFDD (Ours)** | **2.03×** | **2.05** | **2.09×** | **1.99** | **1.81×** | **1.81** | **1.54×** | **1.55** | **1.66×** | **1.54** | **1.82×** | **1.79** |
| 10% | Random | 2.25× | 2.49 | 2.21× | 2.24 | 2.04× | 2.09 | 1.80× | 1.84 | 1.73× | 1.76 | 2.01× | 2.08 |
| | **SFDD (Ours)** | **2.32×** | **2.59** | **2.25×** | **2.30** | **2.08×** | **2.13** | **1.93×** | **1.87** | **1.79×** | **1.83** | **2.07×** | **2.14** |
| 20% | Random | 2.27× | 2.72 | 2.34× | 2.43 | 2.08× | 2.35 | 1.83× | 2.14 | 1.81× | 1.90 | 2.06× | 2.31 |
| | **SFDD (Ours)** | **2.38×** | **2.77** | **2.51×** | **2.52** | **2.28×** | **2.40** | **2.02×** | **2.19** | **1.97×** | **2.04** | **2.23×** | **2.39** |

## 5.4 ANALYSIS OF TRAINING EFFICIENCY

A primary motivation for data selection is to improve training efficiency. We therefore examine the training time. All reported times are end-to-end and include the one-off data-selection overhead (random filtering: 0.02s; SFDD: 2242s); see Appendix D. As illustrated in Figure 4, training time scales approximately linearly with the data retain ratio, aligning with the intuition that training speedup is directly proportional to the data reduction rate. Meanwhile, even with a larger selection cost, the SFDD curve lies below the random-filtering curve across retain ratios,

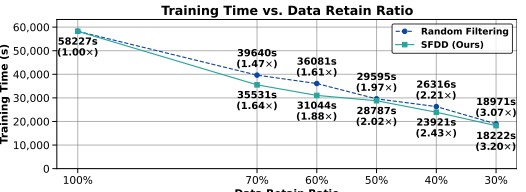

Figure 4: Training time as a function of the data retain ratio (including data-selection time). Each point is annotated with the absolute wall-clock time and the corresponding training speedup.

indicating an improvement in net training speed relative to random filtering. We hypothesize that this arises from enhanced batching efficiency when training on samples with more flat tokens. Moreover, for instance, at the 50% retain ratio, our SFDD method reduces training time from 58,227s (full dataset) to 28,787s, achieving a **2.02× training speedup**, while the inference speedup decreases by less than 4% (Section 5.2). This finding underscores the potential of our approach to substantially reduce computational costs with minimal impact on the final model's SD inference time.

## 6 CONCLUSION

In this work, we address the problem of inefficient draft model training in speculative decoding. We introduce flatness, a novel concept that identifies valuable training data by measuring the target model's predictive uncertainty. We propose a data-centric selection method SFDD, which uses this principle to curate smaller, more potent training datasets. This establishes a new paradigm that significantly enhances training efficiency while preserving the model's inference capabilities. Future work could generalize this approach to other architectures or explore dynamic selection strategies.

## ACKNOWLEDGEMENT

This work is supported by Jiangsu Province Carbon Peak Carbon Neutrality Science and Technology Innovation Special Fund Project (Grant No. BT2025029) and National Natural Science Foundation of China (62576091). Additional support was provided by the Southeast University Big Data Computing Center and the Southeast University Kunpeng & Ascend Center of Cultivation.

## ETHICS STATEMENT

Our research is dedicated to advancing the computational efficiency of training algorithms for speculative decoding. All experiments are conducted using publicly available models and datasets, ensuring transparency and accessibility. We have carefully considered the potential impacts of our work and do not foresee any direct negative societal consequences or ethical concerns. Our methodology is designed to reduce the computational resources required for training large models, which we believe constitutes a positive contribution to the field by promoting more environmentally sustainable and accessible research.

## REPRODUCIBILITY STATEMENT

To ensure full reproducibility, the source code for our method has been made publicly available at `https://github.com/fjm9933/Flatness`. Our experimental setup, including the specific models, datasets, is detailed in Section 5.1. Furthermore, complete theoretical derivations for our proposed approach are provided in Appendix A and Appendix B, allowing for thorough verification of our analytical results. We are committed to transparency and have provided all necessary components for the research community to replicate and build upon our findings.

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

## A   DERIVATION OF THE OPTIMAL GAUSSIAN SOLUTION

In this section, we provide a detailed derivation for the parameters of the optimal distribution $r^*$ under the Gaussian assumption.

**Definition A.1** (Problem Formulation). Let $p = \mathcal{N}(\mu_p, \sigma_p^2)$ and $q = \mathcal{N}(\mu_q, \sigma_q^2)$ be the target and anchor distributions, respectively. We seek an optimal distribution $r^* = \mathcal{N}(\mu_r^*, \sigma_r^{2*})$ that solves the following constrained optimization problem:

$$\min_r \quad D_{\mathrm{KL}}(p\|r) \tag{9}$$

$$\text{subject to} \quad D_{\mathrm{KL}}(r\|q) \leq \theta \tag{10}$$

where $\theta \geq 0$ is a fixed budget. For the sake of simplicity, below we restrict the search space for $r$ to the family of Gaussian distributions.

**Theorem A.2** (Optimal Gaussian Parameters). *There exists a one-parameter family of Gaussian candidates*

$$r(\tau) := \mathcal{N}(\mu_r(\tau), \sigma_r^2(\tau)), \qquad \tau \in [0, 1], \tag{11}$$

*and a unique parameter $\tau^* \in [0, 1]$ such that the optimal solution $r^*$ to the problem above is given by*

$$r^* = r(\tau^*) = \mathcal{N}(\mu_r^*, \sigma_r^{2*}). \tag{12}$$

*For any $\tau \in [0, 1]$, we have*

$$\mu_r(\tau) = (1 - \tau)\,\mu_p + \tau\,\mu_q, \tag{13}$$

$$\sigma_r^2(\tau) = (1 - \tau)\,\sigma_p^2 + \tau\,\sigma_q^2 + \tau^2(1 - \tau)\,(\mu_p - \mu_q)^2. \tag{14}$$

*In particular, $\mu_r^* = \mu_r(\tau^*)$ and $\sigma_r^{2*} = \sigma_r^2(\tau^*)$. Let $\Delta_\mu := \mu_p - \mu_q$ and define*

$$g(\tau) := D_{\mathrm{KL}}(r(\tau)\|q) = \frac{1}{2}\left[\log\frac{\sigma_q^2}{\sigma_r^2(\tau)} + \frac{\sigma_r^2(\tau) + (1 - \tau)^2\Delta_\mu^2}{\sigma_q^2} - 1\right]. \tag{15}$$

*Then:*

- *If $0 \leq \theta < D_{\mathrm{KL}}(p\|q)$, the inequality constraint is active and $\tau^*$ is uniquely determined by $g(\tau^*) = \theta$, yielding $r^* = r(\tau^*)$.*

- *If $\theta \geq D_{\mathrm{KL}}(p\|q)$, the constraint is inactive and $\tau^* = 0$, hence $r^* = p$.*

- *Boundary cases: $\theta = 0 \Rightarrow \tau^* = 1$ (so $r^* = q$); $\theta = D_{\mathrm{KL}}(p\|q) \Rightarrow \tau^* = 0$ (so $r^* = p$).*

*Moreover, if $p \neq q$, the function $g(\tau)$ is strictly decreasing on $\tau \in (0, 1)$, hence the solution $\tau^*$ to $g(\tau) = \theta$ is unique whenever $\theta \in [0, D_{\mathrm{KL}}(p\|q))$.*

*Proof.* We use the method of Lagrange multipliers (KKT for inequality constraints). For one-dimensional Gaussians,

$$D_{\mathrm{KL}}\big(\mathcal{N}(\mu_1, \sigma_1^2)\,\big\|\,\mathcal{N}(\mu_2, \sigma_2^2)\big) = \frac{1}{2}\left[\log\frac{\sigma_2^2}{\sigma_1^2} + \frac{\sigma_1^2 + (\mu_1 - \mu_2)^2}{\sigma_2^2} - 1\right]. \tag{16}$$

Write the decision variables as $(\mu_r, \sigma_r^2)$ with $\sigma_r^2 > 0$, and define

$$J(\mu_r, \sigma_r^2) := D_{\mathrm{KL}}(p\|r) = \frac{1}{2}\left[\log\frac{\sigma_r^2}{\sigma_p^2} + \frac{\sigma_p^2 + (\mu_p - \mu_r)^2}{\sigma_r^2} - 1\right], \tag{17}$$

$$C(\mu_r, \sigma_r^2) := D_{\mathrm{KL}}(r\|q) = \frac{1}{2}\left[\log\frac{\sigma_q^2}{\sigma_r^2} + \frac{\sigma_r^2 + (\mu_r - \mu_q)^2}{\sigma_q^2} - 1\right]. \tag{18}$$

The Lagrangian is

$$\mathcal{L}(\mu_r, \sigma_r^2, \nu) = J(\mu_r, \sigma_r^2) + \nu\big(C(\mu_r, \sigma_r^2) - \theta\big), \qquad \nu \geq 0, \tag{19}$$

with the complementarity condition $\nu\big(C(\mu_r, \sigma_r^2) - \theta\big) = 0$ and primal feasibility $C(\mu_r, \sigma_r^2) \leq \theta$.

**Stationarity conditions.** Taking derivatives,

$$\frac{\partial \mathcal{L}}{\partial \mu_r} = \frac{\mu_r - \mu_p}{\sigma_r^2} + \nu \frac{\mu_r - \mu_q}{\sigma_q^2} = 0, \tag{20}$$

$$\frac{\partial \mathcal{L}}{\partial \sigma_r^2} = \frac{1}{2}\left( \frac{1}{\sigma_r^2} - \frac{\sigma_p^2 + (\mu_p - \mu_r)^2}{(\sigma_r^2)^2} \right) + \nu \frac{1}{2}\left( -\frac{1}{\sigma_r^2} + \frac{1}{\sigma_q^2} \right) = 0. \tag{21}$$

**Step 1: Mean parameter.** From Equation 20,

$$(\mu_r - \mu_p)\,\sigma_q^2 + \nu\,\sigma_r^2(\mu_r - \mu_q) = 0 \;\Rightarrow\; \mu_r = \frac{\sigma_q^2\,\mu_p + \nu\,\sigma_r^2\,\mu_q}{\sigma_q^2 + \nu\,\sigma_r^2}. \tag{22}$$

Introduce the reparameterization

$$\tau := \frac{\nu\,\sigma_r^2}{\sigma_q^2 + \nu\,\sigma_r^2} \in [0,1], \tag{23}$$

which yields the affine interpolation:

$$\mu_r = (1-\tau)\mu_p + \tau\mu_q, \qquad \mu_r - \mu_p = -\tau\Delta_\mu, \quad \mu_r - \mu_q = (1-\tau)\Delta_\mu. \tag{24}$$

If the constraint is inactive ($\nu = 0$), then $\tau = 0$ and $\mu_r = \mu_p$.

**Step 2: Variance parameter.** Using Equation 21, substituting $(\mu_p - \mu_r)^2 = \tau^2\Delta_\mu^2$, and employing $\nu = \dfrac{\tau\,\sigma_q^2}{\sigma_r^2(1-\tau)}$ (from the definition of $\tau$), we obtain

$$\frac{1}{\sigma_r^2} - \frac{\sigma_p^2 + \tau^2\Delta_\mu^2}{(\sigma_r^2)^2} + \frac{\tau}{1-\tau}\left( -\frac{1}{\sigma_r^2} + \frac{1}{\sigma_q^2} \right) = 0. \tag{25}$$

Multiplying by $(\sigma_r^2)^2$ and rearranging gives

$$\frac{\sigma_r^2}{1-\tau} - \frac{\tau\,\sigma_q^2}{1-\tau} - \sigma_p^2 - \tau^2\Delta_\mu^2 = 0, \tag{26}$$

hence the closed form

$$\sigma_r^2(\tau) = (1-\tau)\,\sigma_p^2 + \tau\,\sigma_q^2 + \tau^2(1-\tau)\,\Delta_\mu^2, \tag{27}$$

which proves Equation 14. If the constraint is inactive ($\tau = 0$), then $\sigma_r^2(0) = \sigma_p^2$.

**Step 3: Constraint, monotonicity, and cases.** Insert $\mu_r(\tau)$ and $\sigma_r^2(\tau)$ into $C$ to obtain the scalar function

$$g(\tau) = \frac{1}{2}\left[ \log \frac{\sigma_q^2}{\sigma_r^2(\tau)} + \frac{\sigma_r^2(\tau) + (1-\tau)^2\Delta_\mu^2}{\sigma_q^2} - 1 \right]. \tag{28}$$

Differentiating and using

$$\frac{d}{d\tau}\sigma_r^2(\tau) = -\sigma_p^2 + \sigma_q^2 + \tau(2-3\tau)\Delta_\mu^2, \quad \mu_r(\tau) - \mu_q = (1-\tau)\Delta_\mu, \tag{29}$$

we get

$$g'(\tau) = \frac{1}{2}\left( \frac{1}{\sigma_q^2} - \frac{1}{\sigma_r^2(\tau)} \right)\frac{d}{d\tau}\sigma_r^2(\tau) - \frac{(1-\tau)\Delta_\mu^2}{\sigma_q^2} \;\leq\; 0, \tag{30}$$

with equality only in degenerate cases (e.g. $\tau \in \{0,1\}$ or $\Delta_\mu = 0$ together with $\sigma_p^2 = \sigma_q^2$). Hence $g$ is strictly decreasing on $(0,1)$ whenever $p \neq q$.

*Active-constraint case:* If $0 \leq \theta < D_{\mathrm{KL}}(p\|q)$, complementarity implies $\nu > 0$, thus $\tau \in (0,1]$ and the unique optimal parameter $\tau^*$ satisfies $g(\tau^*) = \theta$.

*Inactive-constraint case:* If $\theta \geq D_{\mathrm{KL}}(p\|q)$, taking $r = p$ yields feasibility $D_{\mathrm{KL}}(p\|q) \leq \theta$ and zero objective value; optimality follows since $J(\cdot) \geq 0$ with equality only at $r = p$. Therefore $\nu = 0$, $\tau^* = 0$, and $r^* = p$.

The boundary $\theta = 0$ gives $\tau^* = 1$ and $r^* = q$; $\theta = D_{\mathrm{KL}}(p\|q)$ gives $\tau^* = 0$ and $r^* = p$. $\qquad\square$

## B  ASYMPTOTIC BEHAVIOR OF THE COSINE SIMILARITY FOR A DISCRETIZED GAUSSIAN

This appendix provides a rigorous derivation of the asymptotic relationship between the cosine similarity and the standard deviation $\sigma$ of a Gaussian distribution as its discretization becomes infinitely fine. Throughout, the window $[-L, L]$ is *large but finite*: large so that the truncated Gaussian approximates the full Gaussian well, and finite so that the discrete uniform distribution is well-defined. This "large window" condition is naturally met in applications like Large Language Models, where predictive distributions over vast vocabularies are highly concentrated.

**Definition B.1** (Discretized Gaussian)**.** Let a discrete probability distribution $p$ over a vocabulary of size $V$ be a discretization of a continuous Gaussian probability density function (PDF) $\phi(x; \mu, \sigma)$ over a symmetric interval $[-L, L]$. The probability $p_i$ in the $i$-th bin of width $\Delta x = 2L/V$ is defined by the PDF at the bin's center $x_i$, such that $p_i = \phi(x_i; \mu, \sigma) \, \Delta x$. This definition ensures $\sum_{i=1}^{V} p_i \to \int_{-L}^{L} \phi(x; \mu, \sigma) \, dx$ as $V \to \infty$. Let $U$ be the discrete uniform distribution $(1/V, \ldots, 1/V)$.

**Theorem B.2** (Asymptotic Cosine Similarity)**.** *In the limit as the vocabulary size $V \to \infty$ (with $L$ fixed and large but finite), the cosine similarity between the discretized Gaussian $p$ and the uniform distribution $U$ satisfies*

$$\lim_{V \to \infty} \cos(p, U) \; = \; \frac{\displaystyle\int_{-L}^{L} \phi(x; \mu, \sigma) \, dx}{\sqrt{2L \displaystyle\int_{-L}^{L} \phi(x; \mu, \sigma)^2 \, dx}}. \tag{31}$$

*Assuming the interval $[-L, L]$ is large enough to contain most of the probability mass, we further obtain the closed-form dependence on $\sigma$:*

$$\lim_{V \to \infty} \cos(p, U) \; = \; \sqrt{\frac{\sigma \sqrt{\pi}}{L}}, \tag{32}$$

*i.e., for fixed (large but finite) $L$, the cosine similarity obeys $\cos(p, U) \propto \sigma^{1/2}$.*

*Proof.* We begin with the cosine similarity written purely in terms of sums:

$$\cos(p, U) \; = \; \frac{\sum_{i=1}^{V} p_i \cdot (1/V)}{\sqrt{\left(\sum_{i=1}^{V} p_i^2\right) \left(\sum_{i=1}^{V} (1/V)^2\right)}} \; = \; \frac{\sum_{i=1}^{V} p_i}{\sqrt{V \sum_{i=1}^{V} p_i^2}}. \tag{33}$$

Define

$$S_V \; := \; \sum_{i=1}^{V} p_i, \qquad Q_V \; := \; V \sum_{i=1}^{V} p_i^2. \tag{34}$$

By the discretization rule $p_i = \phi(x_i; \mu, \sigma) \, \Delta x$ with $\Delta x = 2L/V$, these sums become Riemann sums. In the limit $V \to \infty$, they converge to integrals:

$$S_V \xrightarrow[V \to \infty]{} \int_{-L}^{L} \phi(x; \mu, \sigma) \, dx, \tag{35}$$

and

$$Q_V \; = \; V(\Delta x)^2 \sum_{i=1}^{V} \phi(x_i; \mu, \sigma)^2 \; = \; (2L) \left(\Delta x \sum_{i=1}^{V} \phi(x_i; \mu, \sigma)^2\right) \xrightarrow[V \to \infty]{} 2L \int_{-L}^{L} \phi(x; \mu, \sigma)^2 \, dx. \tag{36}$$

Combining these limits yields the exact fixed-$L$ formula Equation 31.

To make the $\sigma$-dependence explicit, we now invoke a condition that is well-justified in practice. The output of systems like Large Language Models (LLMs) over their vast vocabularies is empirically a long-tail distribution: probability mass is highly concentrated on a few likely outcomes. The Gaussian PDF serves here as a tractable mathematical model for this concentration. A highly

peaked distribution is modeled by a **small standard deviation** $\sigma$. Given that the vocabulary space (represented by $L$) is large, the condition $L \gg \sigma$ is naturally fulfilled.

Under this $L \gg \sigma$ condition, the probability mass in the tails outside $[-L, L]$ is negligible. We can therefore approximate the truncated integrals by their values on the full real line $\mathbb{R}$:

$$\int_{\mathbb{R}} \phi(x; \mu, \sigma)\, dx = 1, \qquad \int_{\mathbb{R}} \phi(x; \mu, \sigma)^2\, dx = \frac{1}{2\sigma\sqrt{\pi}}. \tag{37}$$

This gives the approximation:

$$\int_{-L}^{L} \phi(x; \mu, \sigma)\, dx \approx 1, \qquad \int_{-L}^{L} \phi(x; \mu, \sigma)^2\, dx \approx \frac{1}{2\sigma\sqrt{\pi}}. \tag{38}$$

Substituting Equation 38 into the exact formula Equation 31 gives the final asymptotic result:

$$\lim_{V \to \infty} \cos(p, U) \approx \frac{1}{\sqrt{2L \cdot \frac{1}{2\sigma\sqrt{\pi}}}} = \sqrt{\frac{\sigma\sqrt{\pi}}{L}}, \tag{39}$$

which is Equation 32. This completes the proof. $\qquad\square$

## C  RESULTS AT TEMPERATURE=0

We also evaluate our proposed SFDD when the default temperature is set to 0. The results are summarized in Table 4 and Table 5, which again show that our proposed SFDD can deliver more reliable data selection performance towards more efficient draft model training in the context of SD.

Table 4: Comparison of various metrics for data importance at a 50% retain ratio (temperature = 0).

| Method | GSM8K Speedup | $l$ | Alpaca Speedup | $l$ | MTB Speedup | $l$ | CNN/DM Speedup | $l$ | NQ Speedup | $l$ | Average Speedup | $l$ |
|---|---|---|---|---|---|---|---|---|---|---|---|---|
| No Filter | 2.86× | 3.42 | 2.99× | 3.12 | 3.06× | 3.17 | 2.66× | 2.75 | 2.43× | 2.53 | 2.80× | 3.00 |
| Random | 2.52× | 3.03 | 2.69× | 2.81 | 2.53× | 2.82 | 2.22× | 2.50 | 2.08× | 2.31 | 2.41× | 2.69 |
| Entropy | 2.66× | 2.96 | 2.73× | 2.83 | 2.54× | 2.82 | 2.24× | 2.52 | 2.19× | 2.34 | 2.47× | 2.69 |
| Top-1 Probability | 2.58× | 3.00 | 2.68× | 2.66 | 2.56× | 2.81 | 2.29× | 2.51 | 2.11× | 2.32 | 2.44× | 2.66 |
| Margin | 2.65× | 2.99 | 2.60× | 2.80 | 2.49× | 2.69 | 2.23× | 2.41 | 2.06× | 2.18 | 2.41× | 2.61 |
| Energy Score | 2.52× | 2.91 | 2.69× | 2.80 | 2.57× | 2.79 | 2.24× | 2.50 | 2.15× | 2.29 | 2.43× | 2.66 |
| PPL | 2.54× | 2.91 | 2.69× | 2.90 | 2.49× | 2.80 | 2.25× | 2.51 | 2.22× | 2.32 | 2.44× | 2.69 |
| **SFDD (Ours)** | **2.79×** | **3.07** | **2.92×** | **2.90** | **2.70×** | **2.88** | **2.46×** | **2.54** | **2.24×** | **2.38** | **2.62×** | **2.75** |

Table 5: Comparison of SFDD and Random Filtering under different retain ratios (temperature = 0).

| Retain Ratio | Method | GSM8K Speedup | $l$ | Alpaca Speedup | $l$ | MTB Speedup | $l$ | CNN/DM Speedup | $l$ | NQ Speedup | $l$ | Average Speedup | $l$ |
|---|---|---|---|---|---|---|---|---|---|---|---|---|---|
| 100% | No Filter | 2.86× | 3.42 | 2.99× | 3.12 | 3.06× | 3.17 | 2.66× | 2.75 | 2.43× | 2.53 | 2.80× | 3.00 |
| 70% | Random | 2.58× | 3.01 | 2.70× | 2.79 | 2.50× | 2.79 | 2.21× | 2.51 | 2.08× | 2.30 | 2.41× | 2.68 |
| | **SFDD (Ours)** | **2.79×** | **3.07** | **2.96×** | **2.88** | **2.98×** | **2.89** | **2.43×** | **2.63** | **2.35×** | **2.39** | **2.70×** | **2.77** |
| 60% | Random | 2.60× | 3.01 | 2.71× | 2.78 | 2.59× | 2.82 | 2.27× | 2.50 | 2.15× | 2.30 | 2.46× | 2.68 |
| | **SFDD (Ours)** | **2.70×** | **3.08** | **2.96×** | **2.91** | **2.76×** | **2.87** | **2.43×** | **2.59** | **2.44×** | **2.40** | **2.66×** | **2.77** |
| 50% | Random | 2.52× | 3.03 | 2.69× | 2.81 | 2.53× | 2.82 | 2.22× | 2.50 | 2.08× | 2.31 | 2.41× | 2.69 |
| | **SFDD (Ours)** | **2.79×** | **3.07** | **2.92×** | **2.90** | **2.70×** | **2.88** | **2.46×** | **2.54** | **2.24×** | **2.38** | **2.62×** | **2.75** |
| 40% | Random | 2.56× | 2.97 | 2.54× | 2.74 | 2.56× | 2.82 | 2.24× | 2.50 | 2.00× | 2.26 | 2.38× | 2.66 |
| | **SFDD (Ours)** | **2.66×** | **3.04** | **2.78×** | **2.87** | **2.87×** | **2.82** | **2.45×** | **2.59** | **2.28×** | **2.36** | **2.61×** | **2.74** |
| 30% | Random | 2.50× | 2.92 | 2.60× | 2.72 | 2.50× | 2.73 | 2.23× | 2.40 | 2.05× | 2.23 | 2.38× | 2.60 |
| | **SFDD (Ours)** | **2.59×** | **2.98** | **2.81×** | **2.81** | **2.59×** | **2.79** | **2.37×** | **2.47** | **2.27×** | **2.30** | **2.53×** | **2.67** |

## D  TIMING PROTOCOL AND DATA SELECTION OVERHEAD

**End-to-end timing.**  For fair comparisons, all results related to training time reported in the main text are end-to-end wall-clock measurements that include the data selection stage.

**Cost analysis of SFDD-based data selection.** We note that the cost of SFDD-based data selection is negligible compared to the training cost. In our experiments, we only need to run SFDD-based

data selection once over the training set, which takes only 2,242 seconds and accounts for about 3.85% of the 58,227-seconds whole training time of no filtering. For fair comparisons, our reported training speedups have already included this one-off data selection cost. From the computational structure, this is also expected: during training, each sample requires a forward pass of the target model, a forward pass of the draft model, and a full backward pass, repeated over many epochs. In contrast, the SFDD-based data selection runs only a single forward pass of the target model over the training set. Therefore, the one-off cost of SFDD-based data selection remains negligible compared to multi-epoch training and does not become a practical bottleneck in time or computation.

## E    MORE TRAINING DETAILS

We follow the official EAGLE-2 training setup and fix the target model during all experiments. The pretrained target model is LLaMA3-Instruct-8B. The draft model is a lightweight LLaMA-style predictor with a single transformer layer, initialized and configured according to the official EAGLE-2. The training hyperparameters for the draft model are summarized in Table 6.

Table 6: Training hyperparameters.

| Hyperparameter | Value |
|---|---|
| Number of epochs | 30 |
| Learning rate | $5 \times 10^{-5}$ |
| Batch size | 2 |
| Gradient accumulation steps | 1 |
| Total training steps | 800,000 |
| Warmup | enabled |
| Warmup steps | 2,000 |
| Optimizer | AdamW ($\beta_1 = 0.9$, $\beta_2 = 0.95$) |
| Gradient clipping | 0.5 |
| Maximum sequence length | 2,048 |
| Number of data loader workers | 8 |

## F    ADDITIONAL ANALYSIS

### F.1    ANALYSIS OF INCONSISTENCY BETWEEN ACCEPTANCE LENGTH AND SPEEDUP

In this section, we address the observation from Table 1 regarding the non-linear relationship between acceptance length and inference speedup (e.g., a larger acceptance length gap does not always translate to a proportional speedup gap). We confirm that this phenomenon is not due to hardware mismatch or redundancy but is an inherent characteristic of speculative decoding, consistent with findings in prior works (Li et al., 2024c; Weng et al., 2025). For instance, as shown in Table 1 of EAGLE-2 (Li et al., 2024c), the acceptance length of Vicuna-13B (Lookahead method) on Alpaca (4.89) is larger than on MT-bench (4.83), whereas the inference speedup shows the opposite trend. This indicates that acceptance length cannot be equivalently translated into actual speedup.

Table 7: Length-stratified speculative decoding statistics on GSM8K using SFDD. "Top 50%" denotes the subset containing the longest 50% of samples based on prompt length.

| Method | Speedup (Full) | $l$ (Full) | Speedup (Top-50%) | $l$ (Top-50%) |
|---|---|---|---|---|
| SFDD | 2.6856× | 3.9467 | 2.4688× | 3.9506 |

**Empirical explanations.**    The rationale behind this discrepancy is that the acceptance length only captures the behavior of the decoding stage, while ignoring the prefilling stage. During the decoding stage, both the draft and target models can benefit from the KV cache, so the same acceptance length typically results in similar inference costs when the number of decoded tokens is the same. However, during the prefilling stage—where the KV cache cannot be utilized—the inference cost grows exponentially with respect to the prompt length. As a result, prompts with longer sequences often incur significantly higher inference costs, which reduces the achievable speedup even when the acceptance length during the decoding stage is high.

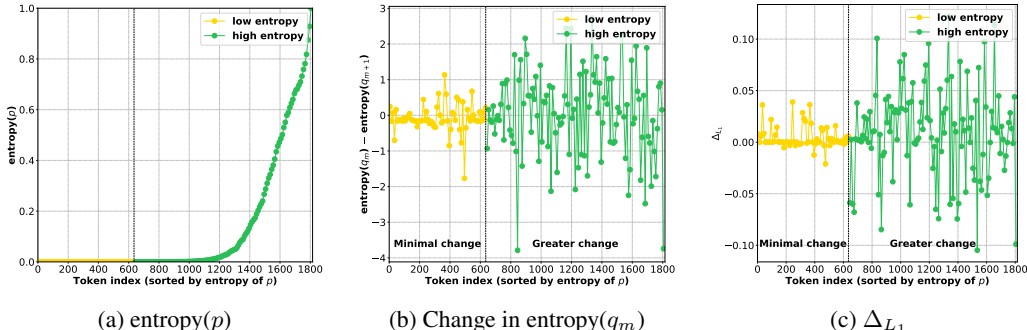

(a) entropy($p$)        (b) Change in entropy($q_m$)        (c) $\Delta_{L_1}$

Figure 5: **Target-sorted entropy view.** Tokens sorted by the target's entropy (low→high entropy). (a) target entropy values; (b) one-epoch change in draft entropy; (c) the one-epoch reduction in the $L_1$ discrepancy, $\Delta_{L_1}$. Similar to the flatness view, we observe that tokens with higher entropy (indicating flatter distributions) contribute more to the training dynamics ($\Delta_{L_1}$) and exhibit larger changes in the draft model.

**Experimental validations.** To substantiate the above explanations, we sort all samples in GSM8K based on their prompt lengths and re-calculate the acceptance length ($l$) and speed on the longest 50% of samples (denoted as "Top 50%"). As shown in Table 7, the speedup on the Top 50% differs from that on the full dataset, even though their acceptance lengths are similar (3.9467 vs. 3.9506). This confirms that prompt length variability may impact the final speedup calculation.

## F.2    ANALYSIS OF ENTROPY AS A DISTRIBUTION-DISPERSION METRIC

In Section 4, we utilize flatness as the primary metric to characterize token distributions. To further validate our findings and verify that the observed trends are not an artifact of the specific metric choice, we perform an additional analysis using entropy. We adopt the identical experimental setup as used for the flatness analysis in Figure 2: we calculate the entropy of the target distribution for all tokens, sort the tokens by their target entropy from low to high, and apply a 10-point moving average for visualization. The results are shown in Figure 5. We observe that the entropy-based curves exhibit a trend remarkably similar to the flatness curves shown in the main text. In the low-entropy region, the draft statistics and $\Delta_{L_1}$ change only slightly within one epoch, whereas in the high-entropy region the changes are more pronounced. This similarity is theoretically expected because both metrics fundamentally measure the distance between the token distribution $p$ and the uniform distribution $U$. While flatness is defined via cosine similarity, entropy is directly related to the forward KL divergence from the uniform distribution (where $U(x) = 1/V$):

$$D_{KL}(p\|U) = \sum_x p(x) \log \frac{p(x)}{1/V} = \sum_x p(x) \log p(x) + \log V = -H(p) + \text{const.} \quad (40)$$

Thus, maximizing entropy is equivalent to minimizing the KL divergence from the uniform distribution. The consistency between Figure 2 (flatness) and Figure 5 (entropy) again confirms that the "flatness" or "uncertainty" of the target distribution is indeed the key factor driving the value of training tokens in speculative decoding.

## F.3    ANALYSIS OF EXPONENTIAL AND HALF-NORMAL DISTRIBUTIONS

In this section, we extend the KL-constrained update from Gaussian to another two distributions: Exponential and Half-normal. We record the core closed-form expressions in our simulations.

**Exponential distribution.** We consider

$$p = \text{Exp}(\lambda_p), \quad q = \text{Exp}(\lambda_q), \quad r = \text{Exp}(\lambda_r), \quad (41)$$

with density $f_\lambda(x) = \lambda e^{-\lambda x}$ on $x \geq 0$. The KL divergence is

$$D_{\text{KL}}\big(\text{Exp}(\lambda_1) \,\|\, \text{Exp}(\lambda_2)\big) = \log \frac{\lambda_1}{\lambda_2} + \frac{\lambda_2}{\lambda_1} - 1. \quad (42)$$

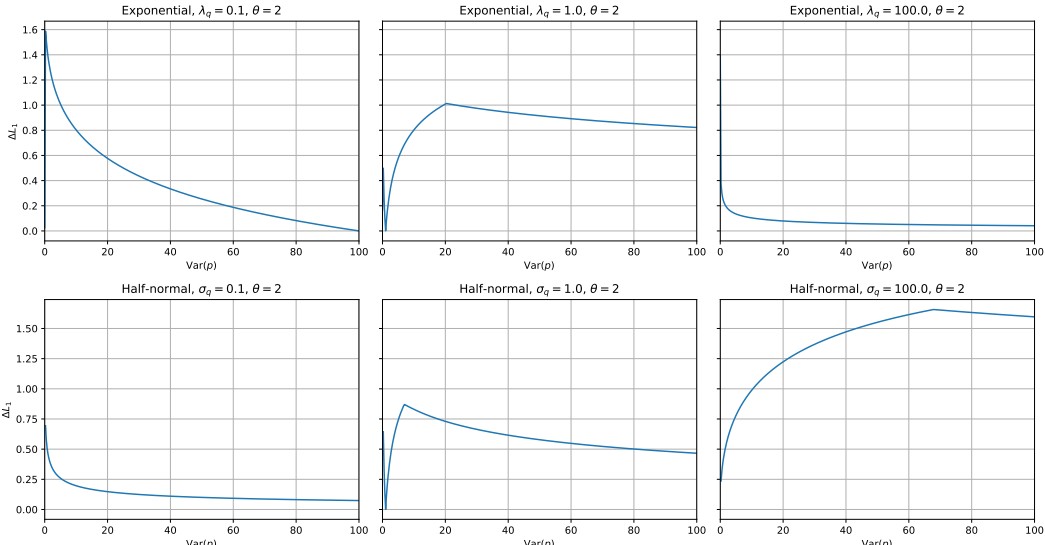

Figure 6: **Behavior of $\Delta L_1$ as a function of the teacher variance in the Exponential and Half-normal families under a large KL budget $\theta = 2$.** Top row: Exponential family with $q = \text{Exp}(\lambda_q)$, $\lambda_q \in \{0.1, 1, 100\}$. Bottom row: Half-normal family with $q = \text{HalfNormal}(\sigma_q)$, $\sigma_q \in \{0.1, 1, 100\}$. In all panels, the $x$-axis is $\text{Var}(p)$ and the $y$-axis is $\Delta L_1 = \|p-q\|_1 - \|p-r^*\|_1$.

For fixed draft parameter $\lambda_q$ and budget $\theta > 0$, the KL ball $\{r : D_{\text{KL}}(r\|q) \leq \theta\}$ induces a closed interval $[\lambda_{\text{low}}, \lambda_{\text{high}}]$ on $\lambda_r$, where $\lambda_{\text{low}}$ and $\lambda_{\text{high}}$ are the two positive solutions of $D_{\text{KL}}(\text{Exp}(\lambda_r)\|\text{Exp}(\lambda_q)) = \theta$, which we solve numerically. The unconstrained minimizer of the update is $\lambda_p$, so the optimal update parameter is simply the projection of $\lambda_p$ onto this interval. The teacher variance in this family is

$$\text{Var}(p) = \frac{1}{\lambda_p^2}. \tag{43}$$

In our simulations, for each variance value we set $\lambda_p = 1/\sqrt{\text{Var}(p)}$, compute the projected $\lambda_r$, and then evaluate $\Delta L_1$ using the closed-form $L_1$ distance between Exponential distributions (omitted here for brevity).

**Half-normal distributions.** We consider

$$p = \text{HalfNormal}(\sigma_p), \quad q = \text{HalfNormal}(\sigma_q), \quad r = \text{HalfNormal}(\sigma_r), \tag{44}$$

with density $f_\sigma(x) = \frac{\sqrt{2}}{\sigma\sqrt{\pi}} e^{-x^2/(2\sigma^2)}$ on $x \geq 0$. Writing $v = \sigma^2$, the KL divergence is

$$D_{\text{KL}}\big(\text{HalfNormal}(\sigma_1) \,\|\, \text{HalfNormal}(\sigma_2)\big) = \frac{1}{2}\Big[\log\frac{v_2}{v_1} + \frac{v_1}{v_2} - 1\Big], \quad v_i = \sigma_i^2. \tag{45}$$

For fixed $v_q = \sigma_q^2$ and budget $\theta$, the KL ball $\{r : D_{\text{KL}}(r\|q) \leq \theta\}$ induces a closed interval $[v_{\text{low}}, v_{\text{high}}]$ on $v_r$, where $v_{\text{low}}$ and $v_{\text{high}}$ are the two positive solutions of $D_{\text{KL}}(\text{HalfNormal}(\sigma_r)\|\text{HalfNormal}(\sigma_q)) = \theta$, again solved numerically. The unconstrained minimizer is $v_p$, so the optimal update variance is the projection of $v_p$ onto $[v_{\text{low}}, v_{\text{high}}]$, and $\sigma_r = \sqrt{v_r}$.

$$\text{Var}(p) = \sigma_p^2 = v_p. \tag{46}$$

In our simulations, for each variance value $\text{Var}(p)$ we set $\sigma_p = \sqrt{\text{Var}(p)}$, we compute the projected $v_r$, and evaluate $\Delta L_1$ using the closed-form $L_1$ distance between Half-normal distributions.

**Numerical behavior of $\Delta L_1$ vs. variance.** Using the formulas above, we numerically evaluate $\Delta L_1(\text{Var}(p))$ under a KL budget $\theta = 2$ for several draft parameters. For the Exponential family, we fix $q = \text{Exp}(\lambda_q)$ with $\lambda_q \in \{0.1, 1, 100\}$ and sweep $\text{Var}(p) \in [0, 100]$. For the Half-normal family, we fix $q = \text{HalfNormal}(\sigma_q)$ with $\sigma_q \in \{0.1, 1, 100\}$ and sweep $\text{Var}(p) = \sigma_p^2 \in [0, 100]$. The

resulting six curves are shown in Figure 6. Across the six panels, the shapes of the $\Delta L_1$–variance curves differ substantially as $\lambda_q$ or $\sigma_q$ varies: the dependence on $\mathrm{Var}(p)$ is highly sensitive to the draft scale, and no simple, consistent monotone trend emerges. In these single-parameter scale families, $p$ and $q$ are forced to share the same mode, so the effect of the teacher's variance on $\Delta L_1$ becomes entangled with the draft's variance and no longer isolates a clean flatness effect, in contrast to the Gaussian location–scale model used in the main text.

**Limitations.** These two families have clear limitations compared with the Gaussian family. First, they are both single-parameter scale families whose mode is fixed at $x = 0$, which forces $p$ and $q$ to share the same maximizer for any choice of parameters. In the LLM setting, this essentially corresponds to the regime where the draft model has already learned the teacher's top-1 token quite well, whereas in speculative decoding we are often more interested in tokens where the teacher and draft still have noticeably different top-1 predictions. Under this "mode-locked" assumption, the effect of the teacher's variance on $\Delta L_1$ becomes strongly entangled with the draft model's variance, and no longer cleanly reflects a flatness effect. In contrast, the Gaussian location–scale family allows us to vary the mean and the variance independently, which better captures the joint effect of argmax mismatch and distributional shape differences observed in real LLM logits. Second, although the KL-constrained optimal update $r^*$ can also be solved for the Exponential and Half-normal cases, it requires solving one-dimensional equations (involving, e.g., Lambert-$W$ functions) and does not admit a simple one-dimensional KKT parameterization as in the Gaussian case, making the expressions less transparent and not revealing any additional phenomena.

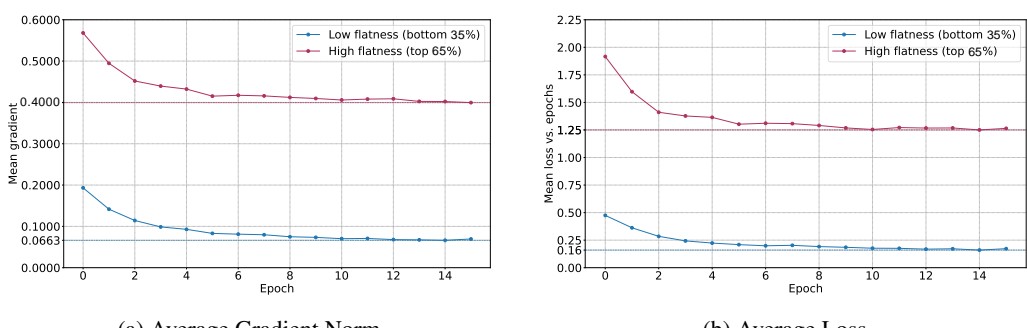

(a) Average Gradient Norm                    (b) Average Loss

Figure 7: Training dynamics of tokens with different flatness levels. We track the metrics for the bottom 35% (Low Flatness) and top 65% (High Flatness) tokens over 15 epochs. **(a) Gradient Norm:** Low-flatness tokens exhibit consistently smaller gradient norms, which quickly decay to negligible values (below 0.1), indicating that the model ceases to learn from them early in training. In contrast, high-flatness tokens maintain significantly larger gradient norms. **(b) Loss Curves:** The loss for low-flatness tokens rapidly converges to near-zero and remains flat, confirming early saturation. Meanwhile, high-flatness tokens maintain non-trivial loss values throughout the epochs, indicating that they continue to provide gradient signals for a longer period.

### F.4  Why we do not use variance of the discrete token distribution

In this section, we further clarify why the variance of the discrete distribution produced by the target LLM is not utilized as a selection criterion in our approach. In the continuous setting, variance relies on the natural order and metric of the real line and therefore has a clear geometric and statistical interpretation. In our setting, however, the language model outputs a categorical distribution over tokens such as "how", "cat", or punctuation marks, which do not live on a canonical one-dimensional numeric axis. If we assign arbitrary indices to tokens and compute a "variance" over these indices, the resulting value would depend entirely on how the vocabulary is indexed: simply permuting the token order would change the variance, so it cannot serve as a robust selection metric. For this reason, when measuring the uncertainty or flatness of LLM outputs, we instead rely on permutation-invariant metrics over the probability vector, such as the cosine-based flatness measure proposed in our paper and other quantities like entropy.

F.5   QUANTITATIVE ANALYSIS OF TOKEN SATURATION

To quantitatively verify the saturation behavior of tokens with different flatness levels, we conduct a tracking experiment. Specifically, we randomly select 10 training samples and sort all constituent tokens by their target flatness. We then split these tokens into two groups: the bottom 35% (representing low-flatness tokens) and the top 65% (representing high-flatness tokens). We track their average gradient norm and loss values from epoch 0 to 15. The results are shown in Figure 7, from which we can have the following three findings: (i) Low-flatness tokens exhibit consistently smaller gradient norms compared to high-flatness tokens across all epochs. Their gradient norms quickly decay to negligible values (below 0.1), indicating that the model ceases to learn from them early in training. (ii) The loss for low-flatness tokens rapidly converges to near-zero and remains flat, confirming early saturation. (iii) In contrast, high-flatness tokens maintain significantly larger gradient norms and non-trivial loss values throughout epochs 0 to 15, indicating that they continue to provide useful gradient signals for a longer period. These results quantitatively support our claim that low-flatness tokens saturate quickly during training, contributing minimal learning signals in later stages, whereas high-flatness tokens continue to drive the optimization process.

F.6   FURTHER ANALYSIS OF TOKEN-LEVEL FILTERING

In the initial phase of this work, we indeed investigate a token-level filtering strategy as a precursor to our sample-level approach. However, under the current training framework, this token-level filtering does not lead to meaningful wall-clock training speedups. The main reasons are two-fold.

- **Forward-pass integrity.** To preserve the necessary autoregressive context for language modeling, the forward pass must be computed over the full sequence. Consequently, masking specific tokens at the loss calculation stage does not obviate the need for their forward computation, yielding no savings in this phase.

- **Negligible savings on the backward pass.** The dominant computational cost in training arises from backpropagation. While token-level loss masking (or clipping) effectively zeroes out gradients at the output layer for masked tokens, standard frameworks must still perform backpropagation through all intermediate Transformer layers over the full sequence length. As a result, the reduction in gradient computation is marginal.

In contrast to token-level filtering, sample-level filtering can discard entire sequences and completely skip both the forward and backward passes for those samples, which is why we ultimately adopt the sample-level SFDD strategy. However, if future training infrastructures allow skipping masked tokens in both forward and backward passes, we believe that flatness-based token-level filtering would become a very promising direction for efficient draft model training.

F.7   THE NECESSITY OF EFFICIENT DRAFT MODEL TRAINING

This section is to clarify that training has become an essential and increasingly non-negligible component in modern speculative decoding (SD) pipelines. While early SD work may rely on a single lightweight SFT step, recent advances have systematically scaled up the training required for competitive acceptance lengths and speedups. As a result, training cost has become more important in realistic deployment scenarios, primarily due to the following two trends:

- **Multi-step SFT.** For example, HASS (Zhang et al., 2024b) and EAGLE-3 (Li et al.) employ more complex training objectives to improve acceptance rates and speedups. These methods are no longer a single lightweight SFT step; instead, they require multi-epoch and heavily supervised training to maximize the achievable acceptance rates and speedups.

- **RL-style training.** For example, GTO (Hu et al., 2025) introduces tree-policy mechanisms and leverages PPO-style optimization, followed by a second-stage RL refinement of the draft model. As reported in its appendix, this refinement incurs substantial training overheads—e.g., 200/400/900 GPU-hours of extra cost for 7B/13B/70B models.

Therefore, as SD training continues to scale up, training efficiency can no longer be treated as negligible. SFDD is designed to be used on top of these methods: it reduces the amount of high-

value training data required to sustain almost the same inference speedups, keeping the additional training cost within a more reasonable budget.

### F.8 ANALYSIS OF UNEXPECTED PERFORMANCE DROP AT 60% RETENTION

We note that, for SFDD, the average acceptance lengths at 50% and 60% retention are almost the same on both MTB and NQ, and the resulting speedups only differ slightly. Because these two retention ratios use very similar amounts of training data, we view such small differences as normal randomness from subset selection and measurement noise, rather than a statistically reliable performance drop. To investigate this, we repeat the experiments on MT-Bench and NQ two additional times under both the 50% and 60% retention settings, and summarize the results in Table 8. We observe that, with nearly identical acceptance lengths, the speedups obtained at 50% and 60% retention remain close across different runs, and the differences between them stay within a small magnitude.

Table 8: Repeated runs of SFDD at 50% and 60% retention on MT-Bench and NQ. The two runs show that speedup and $l$ remain almost the same across repeated experiments.

| Dataset | Retention | Speedup (run 1) ($\times$) | $l$ (run 1) | Speedup (run 2) ($\times$) | $l$ (run 2) |
|---------|-----------|-------------|---------|-------------|---------|
| MT-Bench | 50% | 2.4206 $\times$ | 2.5960 | 2.3947 $\times$ | 2.5960 |
| MT-Bench | 60% | 2.3958 $\times$ | 2.5742 | 2.3927 $\times$ | 2.5742 |
| NQ | 50% | 2.1323 $\times$ | 2.1676 | 2.1352 $\times$ | 2.1676 |
| NQ | 60% | 2.1325 $\times$ | 2.1544 | 2.1345 $\times$ | 2.1544 |

## G  ADDITIONAL EXPERIMENTAL RESULTS

### G.1  EXPERIMENTS BEYOND LLaMA3-8B-INSTRUCT AND SHAREGPT

To further demonstrate the generalization and robustness of our proposed SFDD method, we conduct additional experiments extending beyond the primary LLaMA3-8B-Instruct and ShareGPT. First, we apply SFDD to a different model family, Vicuna-7B-v1.3. The experimental settings for this evaluation are identical to those used in the main paper. Second, to assess robustness on a distinct data distribution, we train the EAGLE draft model (based on LLaMA3-8B-Instruct) from scratch using the GSM8K training split, and evaluate its performance on the test set. The results are summarized in Table 9 and Table 10. The experimental results are basically consistent with the main results in our paper, demonstrating the effectiveness of flatness.

Table 9: Comparison of metrics for data importance on Vicuna-7B-v1.3 at a 50% retain ratio.

| Method | GSM8K | | Alpaca | | MTB | | CNN/DM | | NQ | | Average | |
|--------|---------|---|---------|---|---------|---|---------|---|---------|---|---------|---|
| | Speedup | $l$ | Speedup | $l$ | Speedup | $l$ | Speedup | $l$ | Speedup | $l$ | Speedup | $l$ |
| No Filter | 2.98$\times$ | 3.50 | 2.71$\times$ | 2.95 | 3.14$\times$ | 3.03 | 2.60$\times$ | 2.76 | 2.26$\times$ | 2.22 | 2.74$\times$ | 2.89 |
| Random | 2.83$\times$ | 3.36 | 2.68$\times$ | 2.83 | 2.72$\times$ | 2.97 | 2.40$\times$ | 2.48 | 2.21$\times$ | 2.17 | 2.57$\times$ | 2.76 |
| Entropy | 2.81$\times$ | 3.36 | 2.72$\times$ | 2.90 | 2.83$\times$ | 2.99 | 2.49$\times$ | 2.47 | 2.35$\times$ | 2.31 | 2.64$\times$ | 2.81 |
| Top-1 Probability | 2.69$\times$ | 3.24 | 2.74$\times$ | 2.94 | 2.84$\times$ | 3.00 | 2.57$\times$ | 2.64 | 2.25$\times$ | 2.25 | 2.62$\times$ | 2.81 |
| Margin | 2.65$\times$ | 3.18 | 2.66$\times$ | 2.79 | 2.71$\times$ | 2.87 | 2.53$\times$ | 2.55 | 2.23$\times$ | 2.18 | 2.56$\times$ | 2.71 |
| Energy Score | 2.78$\times$ | 3.34 | 2.76$\times$ | 2.85 | 2.74$\times$ | 2.97 | 2.57$\times$ | 2.64 | 2.27$\times$ | 2.21 | 2.62$\times$ | 2.80 |
| PPL | 2.78$\times$ | 3.35 | 2.68$\times$ | 2.82 | 2.75$\times$ | 2.97 | 2.52$\times$ | 2.57 | 2.28$\times$ | 2.23 | 2.60$\times$ | 2.79 |
| **SFDD (Ours)** | **2.96$\times$** | **3.40** | **2.79$\times$** | **3.04** | **3.16$\times$** | **3.09** | **2.63$\times$** | **2.71** | **2.40$\times$** | **2.34** | **2.79$\times$** | **2.92** |

### G.2  ROBUSTNESS OF SAMPLE-LEVEL AGGREGATION STRATEGIES

In our method, we utilize the arithmetic mean to aggregate token-level flatness scores. To verify robustness, we evaluate an alternative aggregation strategy: the median. We conduct experiments using the same setup as in the main paper (LLaMA3-8B-Instruct, 50% retain ratio). As shown in Table 11, we observe that the performance under median aggregation is very similar to the mean-aggregation results in Table 1. These findings indicate that introducing a robust median-based aggregator does not change our main conclusions and is consistent with our mean-based analysis.

Table 10: Comparison on the GSM8K training set. The results are evaluated on the test set. The EAGLE draft model (based on LLaMA3-8B-Instruct) is trained from scratch on the GSM8K training split at a 50% retain ratio.

| Method | Speedup | $l$ |
|---|---|---|
| No Filter | 1.40× | 1.26 |
| Random | 1.25× | 1.10 |
| Entropy | 1.31× | 1.18 |
| Top-1 Probability | 1.29× | 1.15 |
| Margin | 1.28× | 1.13 |
| Energy Score | 1.31× | 1.18 |
| PPL | 1.31× | 1.17 |
| **SFDD (Ours)** | **1.38×** | **1.24** |

Table 11: Comparison of SFDD and Top-1 Probability at a 50% retain ratio under median aggregation over tokens within each sample; No Filter and Random are also included.

| | GSM8K | | Alpaca | | MTB | | CNN/DM | | NQ | | Average | |
|---|---|---|---|---|---|---|---|---|---|---|---|---|
| Method | Speedup | $l$ | Speedup | $l$ | Speedup | $l$ | Speedup | $l$ | Speedup | $l$ | Speedup | $l$ |
| No Filter | 2.71× | 3.28 | 2.71× | 2.89 | 2.53× | 2.77 | 2.30× | 2.58 | 2.19× | 2.37 | 2.49× | 2.78 |
| Random | 2.43× | 2.85 | 2.37× | 2.59 | 2.26× | 2.48 | 1.99× | 2.31 | 1.93× | 2.06 | 2.20× | 2.46 |
| Top-1 Probability | 2.50× | 2.89 | 2.45× | 2.66 | 2.26× | 2.52 | 2.00× | 2.34 | 1.95× | 2.12 | 2.23× | 2.51 |
| **SFDD (Ours)** | **2.66×** | **2.94** | **2.69×** | **2.73** | **2.47×** | **2.59** | **2.16×** | **2.37** | **2.16×** | **2.18** | **2.43×** | **2.56** |

# H  STATEMENT ON THE USE OF LARGE LANGUAGE MODELS

In this work, we use LLMs to assist with improving the clarity, grammar, and overall readability of the text. All technical content, including theoretical claims and experimental results, is the original work of the authors. The LLMs only serve as a writing aid, and the final manuscript has been carefully reviewed and revised by the authors to ensure its accuracy and originality.

