# OpenReview forum: "Flatter Tokens are More Valuable for Speculative Draft Model Training"
_ICLR.cc/2026/Conference — ICLR 2026 Poster_

### Official Review · Reviewer_tbqx · 2025-10-31

**Soundness:** 3
**Presentation:** 4
**Contribution:** 3
**Rating:** 8
**Confidence:** 4

**Summary:**

This paper reveals that tokens inducing flatter predictive distributions from the target model are more valuable than those yielding sharply peaked distributions. Based on this insight, this work introduces flatness, a new metric to quantify the quality of training tokens, as well as the Sample-level-flatness-based Dataset Distillation (SFDD) approach, which filters the training data to retain only the most valuable samples. Experiments on the EAGLE framework demonstrate that SFDD achieves over 2 training speedup using only 50% of the data, while keeping the final model's inference speedup within 4% of the full-dataset baseline.

**Strengths:**

1. The manuscript is clearly written, with a well-structured narrative, compelling motivation, detailed analyses, and transparent demonstrations that enhance its readability and impact.
2. The motivation of this work is well demonstrated. Existing speculative decoding (SD) approaches such as Eagle series, medusa, and Hydra fine-tunes on all training tokens and treat these tokens equally. Investigating the importance of training tokens to SD is quite an important research question.
3. Through empirical analysis, this work reveals that tokens inducing ≈er predictive distributions from the target model are more valuable than those yielding sharply peaked distributions. The empirical experiments are well designed and clearly demonstrated, with detailed formulations. This research insight may inspire future SD research.
4. The proposed Sample-level-flatness-based Dataset Distillation (SFDD) approach is simple yet effective, which further highlights the utility of the flatness score in practice.

**Weaknesses:**

1. **The necessity of training efficiency**: The major concern of this work is the training efficiency of SD methods. As we know, training efficiency is also a highlighted point in the paper of the Eagle series. In such an SFT setting with only a small amount of training data, the training overhead of SD is quite small, e.g., the training of EAGLE is completed in 1-2 days on 4x A100 (40G) GPUs. The training of EAGLE on 7B, 13B, and 33B models can even be conducted on an RTX 3090 node in 1-2 days. Considering this, the value of inference efficiency improvement may be much higher than the training efficiency of SD.
2. Some errors that could be fixed in the manuscript:
   - Hinton et al. (2015) should be fixed to \citep in Lines 48-49.
   - insert blanks before citation in Lines 84-89.

**Questions:**

Please check the weakness part above. In general, this manuscript is well written with valuable research insights. I recommend the publication of this work in the venue.

---

> ### Author Response · Authors · 2025-11-21
> **Response to Reviewer tbqx (Part 1/1)**
>
> **Dear Reviewer tbqx,**
>
> We sincerely appreciate your constructive feedback and valuable suggestions. In the revised manuscript, we have included thorough discussions to address each of your concerns. Our detailed responses are as follows.
>
> ---
>
> **R4W1: The necessity of training efficiency**
>
> We would like to clarify that training has become an essential and increasingly non-negligible component in modern speculative decoding (SD) pipelines. While early SD work may rely on a single lightweight SFT step, recent advances have systematically scaled up the training required for competitive acceptance lengths and speedups. As a result, training cost has become more important in realistic deployment scenarios, primarily due to the following two trends:
>
> * **Multi-step SFT.** For example, HASS [1] and EAGLE-3 [2] employ more complex training objectives to further improve acceptance lengths and speedups. These methods are no longer a single lightweight SFT step; instead, they require multi-epoch and heavily supervised training to maximize the achievable inference speedups.
> * **RL-style training.** For example, GTO [3] introduces tree-policy mechanisms and leverages PPO-style optimization, followed by a second-stage RL refinement of the draft model. As reported in its appendix, this refinement incurs substantial training overheads — e.g., approximately 200/400/900 GPU-hours of extra cost for 7B/13B/70B models.
>
> Therefore, as SD training continues to scale up, training efficiency can no longer be treated as negligible. SFDD is designed to be used on top of these methods: it reduces the amount of high-value training data required to sustain almost the same inference speedups, while keeping the additional training cost within a more reasonable budget.
>
> We have updated the above in our revised manuscript, which can be found in Appendix F.7.
>
> ---
>
> **R4W2: Formatting issues**
>
> We sincerely apologize for these formatting issues. Following the reviewer's suggestion, we have corrected all issues mentioned and have also thoroughly checked our manuscript to eliminate similar typographical and formatting issues.
>
> ---
>
> **References**
>
> [1] Learning Harmonized Representations for Speculative Sampling.
>
> [2] EAGLE-3: Scaling up Inference Acceleration of Large Language Models via Training-Time Test.
>
> [3] Bridging Draft Policy Misalignment: Group Tree Optimization for Speculative Decoding.

---

### Official Review · Reviewer_9XAi · 2025-11-01

**Soundness:** 3
**Presentation:** 4
**Contribution:** 3
**Rating:** 6
**Confidence:** 4

**Summary:**

This paper finds that in training-based speculative decoding, tokens whose target-model distributions are “flatter” (i.e., closer to uniform) contribute more significantly to improving the acceptance rate. Building on this insight, the authors propose SFDD (Sample-level-flatness-based Dataset Distillation), a data selection strategy that trains the draft model only on samples with high flatness. This approach reduces training cost while preserving inference speedup. Experiments show that SFDD outperforms other data importance metrics such as entropy and perplexity.

**Strengths:**

1. The paper provides a clear, theoretically motivated insight that tokens with flat target distributions are more valuable for SD training, which goes against conventional wisdom in standard KD.
2. The proposed flatness metric is simple, target-model-only, and computable offline. It consistently outperforms multiple baselines across diverse tasks and data retention ratios.
3. SFDD offers a plug-and-play method to significantly reduce training cost for SD without architectural or loss-function changes, which is highly relevant given the growing adoption of SD in LLM serving.

**Weaknesses:**

1. The Gaussian approximation is elegant but may not fully capture the heavy-tailed, sparse nature of real LLM output distributions. The robustness of conclusions to this modeling choice could be better addressed.
2. While flatness is shown to outperform entropy empirically, the paper does not deeply analyze why cosine similarity to uniform is superior to entropy as a flatness proxy since both measure distributional spread.
3. Although SFDD reduces the training cost of the draft model, it requires a full forward pass over the entire training dataset using the target model to compute flatness. When the target model is significantly larger than the draft model, this data selection step may actually introduce additional computational and time overhead.
4. Some experimental settings are unclear, e.g. the pretrained model used for the draft model and the hyperparameters employed during training.
5. The y-axis label in Figure 2(b) is missing a closing parenthesis “)”.

**Questions:**

1. The paper claims that low-flatness tokens “saturate quickly”. Could this be quantified (e.g., by showing their gradient norms or loss reduction curves over training steps)?
2. Why use cosine similarity instead of more commonly used metrics for measuring the discrepancy between two probability distributions, such as KL divergence?
3. The paper adopts a sample-level filtering strategy but does not investigate whether a token-level filtering approach would be effective. Specifically, during training, could masking out low-flatness tokens in the loss computation further reduce computational overhead without degrading performance?

---

> ### Author Response · Authors · 2025-11-21
> **Response to Reviewer 9XAi (Part 1/2)**
>
> **Dear Reviewer 9XAi,**
>
> We sincerely appreciate your constructive feedback and valuable suggestions. In the revised manuscript, we have included extensive experiments and thorough discussions to address each of your concerns. Our detailed responses are as follows.
>
> ---
>
> **R3W1: Discussion of the robustness of conclusions**
>
> We use the 1D Gaussian approximation solely to obtain a clean and interpretable theoretical mechanism: using Gaussian family is a natural theoretical choice and gives a clean KKT analysis, from which we derive the mechanism that flatter targets yield larger single-step reductions in $\Delta L_1$. We then verify this prediction on real LLM outputs: Fig. 2 measures flatness on the discrete target distribution, and the results support the same conclusion across a wide range of tokens. Thus, these empirical result show our takeaway is not tied to the Gaussian approximation and therefore strengthens the robustness of the result.
>
> ---
>
> **R3W2: The reason why cosine similarity outperforms entropy**
>
> We would like to clarify that although both entropy and our flatness metric characterize how dispersed a token distribution is relative to the uniform distribution, they can induce different orderings when used as ranking scores. To substantiate this difference, below we further present an empirical experiment.
>
> **Setup.** We randomly sample $N\in{10,20,30,40,50}$ training examples. For each metric (entropy or flatness), we rank all tokens by that metric and take the bottom 35% as low-score tokens. On the low-score tokens selected by each metric, we compute the average $|\Delta L_1|$ between consecutive training epochs, as a proxy for their remaining impact on SD in late training. We then report the gap $g$ between entropy and flatness.
>
> **Results.** The results are shown in Fig. 2d. We observe $g>0$ consistently (e.g., $3.1\times10^{-4}$ at $N=10$). And the gap increases as $N$ grows, reaching $18.3\times10^{-4}$ at $N=50$. This indicates that flatness-based filtering removes more already-saturated tokens (with smaller $|\Delta L_1|$), explaining why it works better in our data selection.
>
>
> Taken together, these observations give a quantitative explanation of why cosine similarity to the uniform distribution is a better flatness proxy than entropy in our data selection experiments: flatness is more effective at filtering out low-quality tokens that offer minimal training value, leading to higher training efficiency.
>
> We have updated the above in our revised manuscript, which can be found in Section 4.1 (see Fig. 2d).
>
> ---
>
> **R3W3: Overheads of SFDD-based data selection**
>
> We would like to clarify that the cost of SFDD-based data selection is negligible compared to the overall training cost. In our experiments, we only need to run SFDD-based data selection **once** over the training set, which takes only 2,242 seconds and accounts for about 3.85% of the 58,227-seconds whole training time of no filtering. For fair comparisons, our reported training speedups (in Section 5.4) have already included this one-off data selection cost.
>
>
> From the computational structure, this is also expected: during training, each sample requires a forward pass of the target model, a forward pass of the draft model, and a full backward pass, repeated over many epochs. In contrast, the SFDD-based data selection runs only a single forward pass of the target model over the training set. Therefore, the one-off cost of SFDD-based data selection remains negligible compared to multi-epoch training and does not become a practical bottleneck in time or computation.
>
> We have updated the above in our revised manuscript, which can be found in Appendix D.
>
> ---
>
> **R3W4: Unclear experimental settings**
>
> Following the reviewer’s suggestion, we have updated the complete training details in the following table, including the target model, the draft model, and all training-related hyperparameters, which can be found in Appendix E (see Table 6).
>
>
> Table: Training hyperparameters.
>
> | Hyperparameter | Value |
> | :--- | :--- |
> | Number of epochs | 30 |
> | Learning rate | $5\times 10^{-5}$ |
> | Batch size | 2 |
> | Gradient accumulation steps | 1 |
> | Total training steps | 800,000 |
> | Warmup | enabled |
> | Warmup steps | 2,000 |
> | Optimizer | AdamW ($\beta_1=0.9,\ \beta_2=0.95$) |
> | Gradient clipping | 0.5 |
> | Maximum sequence length | 2,048 |
> | Number of data loader workers | 8 |
>
> ---
>
> **R3W5: Labeling mistakes in Figure 2b**
>
> We sincerely apologize for this mistake. Taking the reviewer's comment into account, we have fixed this in our revised manuscript, which can be found in Fig. 2b.

---

> > ### Author Response · Authors · 2025-11-21
> > **Response to Reviewer 9XAi (Part 2/2)**
> >
> > **R3Q1: Quantifying the "saturate quickly" claim**
> >
> > Following the reviewer's suggestion, we have included a quantitative analysis in our revised manuscript. Specifically, we randomly select 10 training samples, sort all tokens by their flatness, split these tokens into the bottom 35% (low flatness) and top 65% (high flatness), and track their average gradient norm and loss reductions from epoch 0 to 15. The results are shown in Fig. 7a (i.e., average gradient norm reductions) and Fig. 7b (i.e., average loss reductions) in Appendix F.6.
> > From these figures, we observe that:
> > - Low-flatness tokens exhibit consistently smaller gradient norms compared to high-flatness tokens across all epochs. Their gradient norms quickly decay to negligible values (below 0.1), indicating that the model ceases to learn from them early in training.
> > - The loss for low-flatness tokens rapidly converges to near-zero and remains flat, confirming early saturation.
> > - In contrast, high-flatness tokens maintain significantly larger gradient norms and non-trivial loss values throughout epochs 0 to 15, indicating that they continue to provide useful gradient signals for a longer period.
> >
> >
> > These results quantitatively support our claim that low-flatness tokens saturate quickly during training, contributing minimal learning signals in later stages, whereas high-flatness tokens continue to drive the optimization process.
> >
> >
> > We have updated the above in our revised manuscript, which can be found in Appendix F.5 (see Fig. 7).
> >
> > ---
> >
> > **R3Q2: The reason why we use cosine similarity instead of KL divergence**
> >
> > In practice, the forward KL divergence from the uniform distribution is equivalent to negative entropy (up to a constant), as shown below:
> > $$
> > D_{KL}(p || U) = \sum_{x} p(x) \log \frac{p(x)}{1/V} = -H(p) + \log V.
> > $$
> > While entropy serves as a standard baseline, our experimental results demonstrate that flatness is a superior metric. In addition to our empirical analysis, as quantified in Fig. 2d, flatness is more effective at filtering out low-quality tokens that offer minimal training value, leading to higher training efficiency.
> >
> > In addition, we have also performed a more fine-grained analysis to compare entropy and flatness (see Fig. 2d). **The reviewer may refer to our response to R3W2 for more details.**
> >
> > ---
> >
> > **R3Q3: Discussion of token-level filtering**
> >
> > We would like to clarify that our work originally starts from a token-level perspective: we first compute a flatness score for each token and then mask low-flatness tokens during training. Empirically, even when we retain only about the top 30% highest-flatness tokens, the speculative decoding behavior at inference time (e.g., acceptance rate and speedup) remains almost unchanged, indicating that token-level flatness is indeed discriminative. However, under the current training framework, this token-level filtering does not lead to meaningful wall-clock training speedups. The main reasons are two-fold:
> >
> > - **Forward-pass integrity.** To preserve the autoregressive context, the forward pass must still be computed on the full sequence; masking some tokens in the loss cannot skip their forward computation.
> > - **Negligible savings on the backward pass.** In practice, the dominant training cost comes from backpropagation. With token-level loss masking or clipping only at the output layer, we may save a small amount of gradient computation in the last layer, but backpropagation through all intermediate Transformer layers still has to be performed over the full sequence length.
> >
> > In contrast to token-level filtering, sample-level filtering can discard entire sequences and completely skip both the forward and backward passes for those samples, which is why we ultimately adopt the sample-level SFDD strategy. However, if future training infrastructures allow skipping masked tokens in both forward and backward passes, we believe that flatness-based token-level filtering would become a very promising direction for efficient draft model training.
> >
> > We have updated the above in our revised manuscript, which can be found in Appendix F.6.

---

### Official Review · Reviewer_w786 · 2025-11-02

**Soundness:** 3
**Presentation:** 3
**Contribution:** 2
**Rating:** 4
**Confidence:** 4

**Summary:**

This paper focuses on improving the training efficiency of drafter model training via dataset selection in a speculative decoding setup. Based on an analysis involving Gaussian distributions, the paper proposes a **flatness** metric to assess the value of a token towards the drafter LM training. The paper then aggregates the per-token flatness values to obtain a sample (sequence ) level metric for the data selection. The paper then performs experiments by using EAGLE-2 training pipeline with LLaMA3-8B-Instruct to showcase the utility of the proposed *Sample-level-flatness-based Dataset Distillation* (SFDD) method towards improving the training efficiency for the drafter model while preserving most of the inference speedup via speculative decoding.

**Strengths:**

- The paper aims to develop a theoretical underpinning for the data selection for draft model training.
- The proposed method, namely SFDD, strikes a good trade-off between training efficiency and inference speedup across multiple datasets. SFDD outperforms other selection criteria from the literature when selecting 50% of the available data from the draft LM training.
- The paper provides ablation studies that highlight the utility of SFDD as one varies the fraction of data selected for the drafter LM training.

**Weaknesses:**

- The theoretical analysis in the paper is based on the assumption that the underlying distributions are Gaussian, which could be far from the discrete distributions produced by an LM. Did the authors consider working with other distributions such as Exponential and Half-normal distribution.
- The empirical evaluation in the paper is a bit limited. The authors may want to expand their empirical study by exploring more LLMs and training datasets.
- The paper does not provide insights on why other natural selection measures such as entropy do not provide good results (see Questions section as well) as compared to the proposed flatness metric. Adding such an analysis would strengthen the contributions of the paper.

**Questions:**

- Figure 1a shows that the reduction in $\Delta L\_1$ is proportional to $\sigma$. Since entropy for the Gaussian is a function of $\sigma$, why did the authors not consider entropy as the metric of interest?
- In Line 252, the paper say ``....we cannot directly compute "continuous variance"``. Did author consider working with variance of the discrete distribution produced by target LM itself as the selection criterion?
- Did you consider other robust aggregation approaches in Eq (8) such as median. Would the conclusions from Table 1 hold with such robust aggregation metrics as well?
- Why did you not include Top-1 probability -- the second best performing baseline in Table 1 -- for your ablation studies in Table 2?
- From Table 1, it appears that the speedup for Random degrades more gently as one decreases the Retain Ration. Did you consider extreme Retain Ratios such 5% and 10%?

---

> ### Author Response · Authors · 2025-11-21
> **Response to Reviewer w786 (Part 1/3)**
>
> **Dear Reviewer w786,**
>
> We sincerely appreciate your constructive feedback and valuable suggestions. In the revised manuscript, we have included extensive experiments and thorough discussions to address each of your concerns. Our detailed responses are as follows.
>
> ---
>
> **R2W1: Working with other distributions (e.g., Exponential and Half-Normal)**
>
> We chose the Gaussian distribution primarily for its mathematical tractability and interpretability. Under the Gaussian assumption, the relationship between the target's variance (flatness) and the training gain ($\Delta L_1$) admits a simple solution. This allows us to explicitly derive how flatness impacts the acceptance rate without resorting to complex numerical approximations.
>
> Following the reviewer's suggestion, we have included simulations for Exponential and Half-Normal distributions in Appendix F.3. However, we note that these distributions typically have a fixed mode at zero. This limitation forces the draft and target distributions to share the same peak location. This fails to capture the critical speculative decoding training scenario where the draft and target disagree. Consequently, the effect of target variance becomes entangled with the draft's variance under these distributions.
>
> We have updated the above in our revised manuscript, which can be found in Appendix F.3 (see Fig. 6).
>
> ---
>
> **R2W2: Experiments beyond LLaMA3-8B-Instruct and ShareGPT**
>
> Following the reviewer's suggestion, we have extended our evaluation to include another representative model family (i.e., Vicuna-7B-v1.3) and dataset (i.e., GSM8K). The results are summarized in the following tables. In both settings, we observe that the performance trends closely match those reported for LLaMA3-8B-Instruct and ShareGPT. This consistency further demonstrates that the effectiveness of SFDD generalizes well across different model architectures and training datasets.
>
>
> Table: Comparison of various metrics for data selection on Vicuna-7B-v1.3 at a 50% retain ratio.
>
> | Method | GSM8K ($\times$) | GSM8K ($l$) | Alpaca ($\times$) | Alpaca ($l$) | MTB ($\times$) | MTB ($l$) | CNN/DM ($\times$) | CNN/DM ($l$) | NQ ($\times$) | NQ ($l$) | Average ($\times$) | Average ($l$) |
> | :--- | :---: | :---: | :---: | :---: | :---: | :---: | :---: | :---: | :---: | :---: | :---: | :---: |
> | No Filter | 2.98 $\times$ | 3.50 | 2.71 $\times$ | 2.95 | 3.14 $\times$ | 3.03 | 2.60 $\times$ | 2.76 | 2.26 $\times$ | 2.22 | 2.74 $\times$ | 2.89 |
> | Random | 2.83 $\times$ | 3.36 | 2.68 $\times$ | 2.83 | 2.72 $\times$ | 2.97 | 2.40 $\times$ | 2.48 | 2.21 $\times$ | 2.17 | 2.57 $\times$ | 2.76 |
> | Entropy | 2.81 $\times$ | 3.36 | 2.72 $\times$ | 2.90 | 2.83 $\times$ | 2.99 | 2.49 $\times$ | 2.47 | 2.35 $\times$ | 2.31 | 2.64 $\times$ | 2.81 |
> | Top-1 Probability | 2.69 $\times$ | 3.24 | 2.74 $\times$ | 2.94 | 2.84 $\times$ | 3.00 | 2.57 $\times$ | 2.64 | 2.25 $\times$ | 2.25 | 2.62 $\times$ | 2.81 |
> | Margin | 2.65 $\times$ | 3.18 | 2.66 $\times$ | 2.79 | 2.71 $\times$ | 2.87 | 2.53 $\times$ | 2.55 | 2.23 $\times$ | 2.18 | 2.56 $\times$ | 2.71 |
> | Energy Score | 2.78 $\times$ | 3.34 | 2.76 $\times$ | 2.85 | 2.74 $\times$ | 2.97 | 2.57 $\times$ | 2.64 | 2.27 $\times$ | 2.21 | 2.62 $\times$ | 2.80 |
> | PPL | 2.78 $\times$ | 3.35 | 2.68 $\times$ | 2.82 | 2.75 $\times$ | 2.97 | 2.52 $\times$ | 2.57 | 2.28 $\times$ | 2.23 | 2.60 $\times$ | 2.79 |
> | **SFDD (Ours)** | **2.96** $\times$ | **3.40** | **2.79** $\times$ | **3.04** | **3.16** $\times$ | **3.09** | **2.63** $\times$ | **2.71** | **2.40** $\times$ | **2.34** | **2.79** $\times$ | **2.92** |
>
>
>
> Table: Comparison on the GSM8K training set. The results are evaluated on the test set. The EAGLE draft model (based on LLaMA3-8B-Instruct) is trained from scratch on the GSM8K training split at a 50% retain ratio.
>
> | Method | Speedup ($\times$) | Average Accepted Length ($l$) |
> | :--- | :---: | :---: |
> | No Filter | 1.40 $\times$ | 1.26 |
> | Random | 1.25 $\times$ | 1.10 |
> | Entropy | 1.31 $\times$ | 1.18 |
> | Top-1 Probability | 1.29 $\times$ | 1.15 |
> | Margin | 1.28 $\times$ | 1.13 |
> | Energy Score | 1.31 $\times$ | 1.18 |
> | PPL | 1.31 $\times$ | 1.17 |
> | **SFDD (Ours)** | **1.38** $\times$ | **1.24** |
>
>
> We have updated the above in our revised manuscript, which can be found in Appendix G.1 (see Table 9 and Table 10).

---

> > ### Author Response · Authors · 2025-11-21
> > **Response to Reviewer w786 (Part 2/3)**
> >
> > **R2W3: The reason why flatness is more reliable than entropy**
> >
> > We would like to clarify that although both entropy and our flatness metric characterize how dispersed a token distribution is relative to the uniform distribution, they can induce different orderings when used as ranking scores. To substantiate this difference, below we further present an empirical experiment.
> >
> > **Settings.** We randomly sample $N\in{10,20,30,40,50}$ training examples. For each metric (entropy or flatness), we rank all tokens by that metric and take the bottom 35% as low-score tokens. On the low-score tokens selected by each metric, we compute the average $|\Delta L_1|$ between consecutive training epochs, as a proxy for their remaining impact on SD in subsequent training. We then report the gap $g$ between entropy and flatness.
> >
> > **Results.** The results are shown in Fig. 2d. We observe $g>0$ consistently (e.g., $3.1\times10^{-4}$ at $N=10$). And the gap increases as $N$ grows, reaching $18.3\times10^{-4}$ at $N=50$. This indicates that, flatness-based filtering removes more already-saturated tokens (with smaller $|\Delta L_1|$), explaining why it works better in our data selection.
> >
> >
> > Taken together, these observations give a quantitative explanation of why cosine similarity to the uniform distribution is a better flatness proxy than entropy in our data selection experiments: flatness is more effective at filtering out low-quality tokens that offer minimal training value, leading to higher training efficiency.
> >
> > We have updated the above in our revised manuscript, which can be found in Section 4.1 (see Fig. 2d).
> >
> >
> > ---
> >
> > **R2Q1: The reason why we do not use entropy as the main metric**
> >
> > In the early stage of this work, we have indeed evaluated entropy as an uncertainty measure, but we have found it less effective than a cosine-similarity-based metric. The experimental results in the paper also show that, under the same retain ratio, flatness-based filtering achieves better SD training performance, with higher acceptance lengths and speedups.
> >
> > **In our response to R2W3**, we further add an empirical analysis, which shows that flatness-based filtering tends to remove more tokens that have already saturated than entropy-based filtering, as shown in Fig. 2d.
> >
> > ---
> >
> > **R2Q2: The reason why we do not use variance of the discrete token distribution**
> >
> > We would like to clarify that we do not use the variance of the discrete distribution produced by the target LLM as a selection criterion, as this quantity does not provide a meaningful notion of uncertainty for categorical variables such as tokens. In the continuous setting, variance relies on the natural order and metric of the real line and therefore has a clear geometric and statistical interpretation. In our setting, however, the language model outputs a categorical distribution over tokens such as "how", "cat", or punctuation marks, which do not live on a canonical one-dimensional numeric axis. If we assign arbitrary indices to tokens and compute a variance over these indices, the resulting value would depend entirely on how the vocabulary is indexed: simply permuting the token order would change the variance, so it cannot serve as a robust selection metric. For this reason, when measuring the uncertainty or flatness of LLM outputs, we instead rely on permutation-invariant metrics over the probability vector, such as the cosine-based flatness measure proposed in our paper and other quantities like entropy.
> >
> > We have updated the above in our revised manuscript, which can be found in Appendix F.4.

---

> > > ### Author Response · Authors · 2025-11-21
> > > **Response to Reviewer w786 (Part 3/3)**
> > >
> > > **R2Q3: Robust aggregation (e.g., median) in Eq. (8)**
> > >
> > > Following the reviewer's suggestion, we have conducted an experiment where we substitute the arithmetic mean in Eq. (8) with the median to aggregate token-level flatness into a sample-level score. The experimental setup remains the same to the main paper. The results are shown in the following table. We observe that the performance under median aggregation is very similar to the mean-aggregation results in Table 1. These findings indicate that introducing a robust median-based aggregator does not change our main conclusions and is consistent with our mean-based analysis.
> > >
> > > Table: Comparison of SFDD and the second-best baseline (Top-1 Probability) at a 50% retain ratio under median aggregation over tokens within each sample.
> > >
> > > | Method | GSM8K ($\times$) | GSM8K ($l$) | Alpaca ($\times$) | Alpaca ($l$) | MTB ($\times$) | MTB ($l$) | CNN/DM ($\times$) | CNN/DM ($l$) | NQ ($\times$) | NQ ($l$) | Average ($\times$) | Average ($l$) |
> > > | :--- | :---: | :---: | :---: | :---: | :---: | :---: | :---: | :---: | :---: | :---: | :---: | :---: |
> > > | No Filter | 2.71 $\times$ | 3.28 | 2.71 $\times$ | 2.89 | 2.53 $\times$ | 2.77 | 2.30 $\times$ | 2.58 | 2.19 $\times$ | 2.37 | 2.49 $\times$ | 2.78 |
> > > | Random | 2.43 $\times$ | 2.85 | 2.37 $\times$ | 2.59 | 2.26 $\times$ | 2.48 | 1.99 $\times$ | 2.31 | 1.93 $\times$ | 2.06 | 2.20 $\times$ | 2.46 |
> > > | Top-1 Probability | 2.50 $\times$ | 2.89 | 2.45 $\times$ | 2.66 | 2.26 $\times$ | 2.52 | 2.00 $\times$ | 2.34 | 1.95 $\times$ | 2.12 | 2.23 $\times$ | 2.51 |
> > > | SFDD (Ours) | 2.66 $\times$ | 2.94 | 2.69 $\times$ | 2.73 | 2.47 $\times$ | 2.59 | 2.16 $\times$ | 2.37 | 2.16 $\times$ | 2.18 | 2.43 $\times$ | 2.56 |
> > >
> > > We have updated the above in our revised manuscript, which can be found in Appendix G.2 (see Table 11).
> > >
> > > ---
> > >
> > > **R2Q4: Include Top-1 probability for ablation study**
> > >
> > > Following the reviewer's suggestion, we have conducted an experiment, where we include Top-1 probability in our ablation study. As shown in Table 2 of our revised manuscript, across all retain ratios from 30% to 70%, SFDD consistently outperforms both Top-1 probability and Random Filtering in terms of average speedup and average acceptance length $l$, further confirming the effectiveness of our flatness-based data selection.
> > >
> > > **We do not put the table here due to the space limitation.**
> > >
> > > We have updated the above in our revised manuscript, which can be found in Section 5.3 (see Table 2).
> > >
> > >
> > > ---
> > >
> > >
> > > **R2Q5: Experiments at extreme retain ratios**
> > >
> > > Following the reviewer's suggestion, we have conducted additional experiments on all five datasets at extreme retain ratios of 5%, 10%, and 20%, comparing Random and SFDD in terms of speedup and average acceptance length. The results are shown in the following table. At very low retain ratios, both methods experience a clear drop in speedup, but SFDD consistently outperforms Random across all datasets. Therefore, Random does not exhibit a “more gentle” degradation in this regime; instead, SFDD maintains consistently better speedup and acceptance, indicating that it remains effective and robust even under extreme retain ratios.
> > >
> > > Table: Ablation study at extreme retain ratios comparing SFDD against Random filtering.
> > >
> > > | Retain Ratio | Method | GSM8K ($\times$) | GSM8K ($l$) | Alpaca ($\times$) | Alpaca ($l$) | MTB ($\times$) | MTB ($l$) | CNN/DM ($\times$) | CNN/DM ($l$) | NQ ($\times$) | NQ ($l$) | Average ($\times$) | Average ($l$) |
> > > | :--- | :--- | :---: | :---: | :---: | :---: | :---: | :---: | :---: | :---: | :---: | :---: | :---: | :---: |
> > > | 5% | Random | 1.75 $\times$ | 1.96 | 2.00 $\times$ | 1.92 | 1.60 $\times$ | 1.73 | 1.48 $\times$ | 1.45 | 1.57 $\times$ | 1.49 | 1.68 $\times$ | 1.71 |
> > > | | **SFDD (Ours)** | **2.03** $\times$ | **2.05** | **2.09** $\times$ | **1.99** | **1.81** $\times$ | **1.81** | **1.54** $\times$ | **1.55** | **1.66** $\times$ | **1.54** | **1.82** $\times$ | **1.79** |
> > > | | | | | | | | | | | | | | |
> > > | 10% | Random | 2.25 $\times$ | 2.49 | 2.21 $\times$ | 2.24 | 2.04 $\times$ | 2.09 | 1.80 $\times$ | 1.84 | 1.73 $\times$ | 1.76 | 2.01 $\times$ | 2.08 |
> > > | | **SFDD (Ours)** | **2.32** $\times$ | **2.59** | **2.25** $\times$ | **2.30** | **2.08** $\times$ | **2.13** | **1.93** $\times$ | **1.87** | **1.79** $\times$ | **1.83** | **2.07** $\times$ | **2.14** |
> > > | | | | | | | | | | | | | | |
> > > | 20% | Random | 2.27 $\times$ | 2.72 | 2.34 $\times$ | 2.43 | 2.08 $\times$ | 2.35 | 1.83 $\times$ | 2.14 | 1.81 $\times$ | 1.90 | 2.06 $\times$ | 2.31 |
> > > | | **SFDD (Ours)** | **2.38** $\times$ | **2.77** | **2.51** $\times$ | **2.52** | **2.28** $\times$ | **2.40** | **2.02** $\times$ | **2.19** | **1.97** $\times$ | **2.04** | **2.23** $\times$ | **2.39** |
> > >
> > >
> > > We have updated the above in our revised manuscript, which can be found in Section 5.4 (see Table 3).

---

### Official Review · Reviewer_wcho · 2025-11-03

**Soundness:** 2
**Presentation:** 2
**Contribution:** 2
**Rating:** 4
**Confidence:** 4

**Summary:**

- The paper introduces a new metric called target model flatness to improve the efficiency of training draft models for speculative decoding.
- Demonstrates that using only 50% of the dataset achieves performance comparable to training on the full dataset.
- Outperforms other baseline methods in experiments.

**Strengths:**

- Novel metric (target model flatness) that could provide better guidance for draft model training.
- Significant data efficiency: achieves similar performance with half the data.
- Shows better results than existing baselines, indicating practical impact.

**Weaknesses:**

- Experiments are limited to Llama3.1-8B-Instruct; unclear if results generalize to other model sizes or families.
- Missing discussion of related work on efficient draft model training (e.g., [1] Goel et al., 2024).
- Lack of clarity on training details (epochs, convergence criteria).

Some tables are confusing:
- Table 1 speed-up discrepancies vs. acceptance length raise concerns about hardware or redundancy:
  - For GSM8K, acceptance lengths for No Filter, SFDD, and PPL are 3.28,2.95,and 2.79 respectively.
  - The gap between No Filter and SFDD is 0.33, while between SFDD and PPL is 0.16.
  - However, the speed-up gap between No Filter and SFDD is only 0.02, while between SFDD and PPL is 0.33.
  - This seems inconsistent: a larger acceptance length difference should not result in a smaller speed-up gap. This seems like a possible hardware mismatch or redundancy for non-SFDD methods.

- Table 2 shows unexpected performance drop at 60% retention for MTB and NQ compared to 50%.
- Mathematical proof (Eq. 5) has ambiguity in computing $\sigma_r^*$​ given its dependence on $\sigma_r$​ through $\tau$.

Reference: [1] Goel, Raghavv, et al. "Direct alignment of draft model for speculative decoding with chat-fine-tuned llms." arXiv preprint arXiv:2403.00858 (2024).

**Questions:**

- Please see weakness section
- Can acceptance plots and target entropy plots be added to Fig. 2? Is there a relationship between entropy and flatness?
- What is the role of acceptance criteria in the top-left of Fig. 3, seems extra?

---

> ### Author Response · Authors · 2025-11-21
> **Response to Reviewer wcho (Part 1/4)**
>
> **Dear Reviewer wcho,**
>
> We sincerely appreciate your constructive feedback and valuable suggestions. In the revised manuscript, we have included extensive experiments and thorough discussions to address each of your concerns. Our detailed responses are as follows.
>
> ---
>
> **R1W1: Experiments beyond LLaMA3-8B-Instruct**
>
> Following the reviewer's suggestion, we have extended our evaluation to include another representative model family (i.e., Vicuna-7B-v1.3). The results are summarized in the following table. We observe that the performance trends closely match those reported for the LLaMA3-8B-Instruct. This consistency further demonstrates that the effectiveness of SFDD generalizes well across different model architectures.
>
> Table: Comparison of various metrics for data selection on Vicuna-7B-v1.3 at a 50% retain ratio.
>
> | Method | GSM8K ($\times$) | GSM8K ($l$) | Alpaca ($\times$) | Alpaca ($l$) | MTB ($\times$) | MTB ($l$) | CNN/DM ($\times$) | CNN/DM ($l$) | NQ ($\times$) | NQ ($l$) | Average ($\times$) | Average ($l$) |
> | :--- | :---: | :---: | :---: | :---: | :---: | :---: | :---: | :---: | :---: | :---: | :---: | :---: |
> | No Filter | 2.98 $\times$ | 3.50 | 2.71 $\times$ | 2.95 | 3.14 $\times$ | 3.03 | 2.60 $\times$ | 2.76 | 2.26 $\times$ | 2.22 | 2.74 $\times$ | 2.89 |
> | Random | 2.83 $\times$ | 3.36 | 2.68 $\times$ | 2.83 | 2.72 $\times$ | 2.97 | 2.40 $\times$ | 2.48 | 2.21 $\times$ | 2.17 | 2.57 $\times$ | 2.76 |
> | Entropy | 2.81 $\times$ | 3.36 | 2.72 $\times$ | 2.90 | 2.83 $\times$ | 2.99 | 2.49 $\times$ | 2.47 | 2.35 $\times$ | 2.31 | 2.64 $\times$ | 2.81 |
> | Top-1 Probability | 2.69 $\times$ | 3.24 | 2.74 $\times$ | 2.94 | 2.84 $\times$ | 3.00 | 2.57 $\times$ | 2.64 | 2.25 $\times$ | 2.25 | 2.62 $\times$ | 2.81 |
> | Margin | 2.65 $\times$ | 3.18 | 2.66 $\times$ | 2.79 | 2.71 $\times$ | 2.87 | 2.53 $\times$ | 2.55 | 2.23 $\times$ | 2.18 | 2.56 $\times$ | 2.71 |
> | Energy Score | 2.78 $\times$ | 3.34 | 2.76 $\times$ | 2.85 | 2.74 $\times$ | 2.97 | 2.57 $\times$ | 2.64 | 2.27 $\times$ | 2.21 | 2.62 $\times$ | 2.80 |
> | PPL | 2.78 $\times$ | 3.35 | 2.68 $\times$ | 2.82 | 2.75 $\times$ | 2.97 | 2.52 $\times$ | 2.57 | 2.28 $\times$ | 2.23 | 2.60 $\times$ | 2.79 |
> | **SFDD (Ours)** | **2.96** $\times$ | **3.40** | **2.79** $\times$ | **3.04** | **3.16** $\times$ | **3.09** | **2.63** $\times$ | **2.71** | **2.40** $\times$ | **2.34** | **2.79** $\times$ | **2.92** |
>
> We have updated the above in our revised manuscript, which can be found in Appendix G.1 (see Table 9).
>
>
> ---
>
> **R1W2: Missing discussion of related work on efficient draft model training**
>
> Following the reviewer’s suggestion, we have added [1] in the related work section, analyzed its key limitation, and clarified the fundamental difference between [1] and our work. Specifically, [1], similar to [2] (as discussed in Section 1 of our paper), also observes that the $L_1$-norm (i.e., total variation distance) is the true theoretical objective of speculative decoding and accordingly proposes a new loss function, TVD++, to facilitate its optimization. However, the objective of [1, 2] is to improve the alignment between the draft and target models, which does not address the challenge of efficient draft model training. In contrast, our work explicitly targets efficient draft model training from the perspective of dataset distillation. To the best of our knowledge, our work is the first to consider efficient draft model training in speculative decoding.
>
> ---
>
> **R1W3: Lack of clarity on training details**
>
> Following the reviewer’s suggestion, we have updated the complete training details in the following table, including the target model, the draft model, and all training-related hyperparameters, which can be found in Appendix E (see Table 6).
>
> Table: Training hyperparameters.
>
> | Hyperparameter | Value |
> | :--- | :--- |
> | Number of epochs | 30 |
> | Learning rate | $5\times 10^{-5}$ |
> | Batch size | 2 |
> | Gradient accumulation steps | 1 |
> | Total training steps | 800,000 |
> | Warmup | enabled |
> | Warmup steps | 2,000 |
> | Optimizer | AdamW ($\beta_1=0.9,\ \beta_2=0.95$) |
> | Gradient clipping | 0.5 |
> | Maximum sequence length | 2,048 |
> | Number of data loader workers | 8 |

---

> ### Author Response · Authors · 2025-11-21
> **Response to Reviewer wcho (Part 2/4)**
>
> **R1W4: Inconsistency between acceptance length and speedup**
>
>
> To clarify, we have carefully checked our experimental settings and results, and we confirm that there is no hardware mismatch or redundancy in our reported results. As shown in several prior works on speculative decoding [3, 4], their reported results also exhibit a similar phenomenon: models with comparable acceptance lengths may still achieve different speedups.
>
> - For example, as shown in Table 1 of [3], the acceptance length of Vicuna-13B (EAGLE-2 method) on Alpaca (4.89) is larger than on MT-bench (4.83), whereas the inference speedup shows the opposite trend. This indicates that acceptance length cannot be equivalently translated into actual speedup.
>
> **Empirical explanations.** The rationale behind this is that the acceptance length only captures the behavior of the decoding stage, while ignoring the prefilling stage. During the decoding stage, both the draft and target models can benefit from KV cache, so the same acceptance length typically results in similar inference cost when the number of decoded tokens is the same. However, during the prefilling stage - where KV cache cannot be used - the inference cost grows exponentially with respect to the prompt length. As a result, prompts with more tokens often incur significantly higher inference cost, which reduces the achievable speedup even when the acceptance length during the decoding stage is the same.
>
> **Experimental validations.** To substantiate the above explanations, we sort all samples in GSM8K based on their prompt lengths and re-calculate the acceptance length and speedup on the longest 50% of samples (denoted as “Top 50%”). As shown in the following table, the speedup on the Top 50% differs from that on the full dataset, even though their acceptance lengths are similar.
>
> Table: Comparison of different prompt lengths on GSM8K. “Top 50%” denotes the longest 50% samples.
>
> | Method | Speedup (Full) | $l$ (Full) | Speedup (Top-50%) | $l$ (Top-50%) |
> | :--- | :---: | :---: | :---: | :---: |
> | SFDD | 2.6856 $\times$ | 3.9467 | 2.4688 $\times$ | 3.9506 |
>
>
> We have updated the above in our revised manuscript, which can be found in Appendix F.1 (see Table 7).
>
> ---
>
>
> **R1W5: Reasons for unexpected performance drop at 60% retention**
>
> We note that, for SFDD, the average acceptance lengths at 50% and 60% retention are almost the same on both MTB and NQ, and the resulting speedups only differ slightly. Because these two retention ratios use very similar amounts of training data, we view such small differences as normal randomness from subset selection and measurement noise, rather than a statistically reliable performance drop.
>
> To further investigate this, we repeat the experiments on MT-Bench and NQ two additional times under both the 50% and 60% retention settings, and summarize the detailed results in the following table. We observe that, with nearly identical acceptance lengths, the speedups obtained at 50% and 60% retention remain consistently close across different runs, and the differences between them stay within a small magnitude.
>
> Table: Repeated runs of SFDD at 50% and 60% retention on MT-Bench and NQ. The two runs (columns “run 1” and “run 2”) show that speedup remains almost the same across repeated experiments.
>
> | Dataset | Retention | Speedup (run 1) ($\times$) | $l$ (run 1) | Speedup (run 2) ($\times$) | $l$ (run 2) |
> | :--- | :--- | :---: | :---: | :---: | :---: |
> | MT-Bench | 50% | 2.4206 $\times$ | 2.5960 | 2.3947 $\times$ | 2.5960 |
> | MT-Bench | 60% | 2.3958 $\times$ | 2.5742 | 2.3927 $\times$ | 2.5742 |
> | NQ | 50% | 2.1323 $\times$ | 2.1676 | 2.1352 $\times$ | 2.1676 |
> | NQ | 60% | 2.1325 $\times$ | 2.1544 | 2.1345 $\times$ | 2.1544 |
>
> We have updated the above in our revised manuscript, which can be found in Appendix F.8 (see Table 8).

---

> ### Author Response · Authors · 2025-11-21
> **Response to Reviewer wcho (Part 3/4)**
>
> **R1W6: Mathematical ambiguity**
>
> We would like to clarify that the root cause of the apparent ambiguity is that we use the same symbol $\tau$ both (i) as the value corresponding to the optimizer that satisfies the budget constraint and (ii) as a generic path parameter on the KKT stationary manifold. These should be properly denoted by $\tau^\*$ and $\tau$, respectively. Under this notation, one might read
> $$
> \tau := \frac{\nu\sigma_r^2}{\sigma_q^2 + \nu\sigma_r^2}
> $$
> as a definition that requires $\sigma_r^2$ to be known before the optimal variance $\sigma_r^{2*}$ is determined, suggesting a circular dependence. In our derivation, however, $\tau$ is introduced only as an *auxiliary parameter* on the KKT stationary manifold: we first express all KKT stationary points as a one-parameter family $(\mu_r(\tau),\sigma_r^2(\tau))$, then determine a unique $\tau^\*$ from the budget constraint $g(\tau^\*)=\theta$, and finally set $\sigma_r^{2*}=\sigma_r^2(\tau^\*)$. At no stage do we compute $\tau$ from $\sigma_r^2$ and plug it back into $\sigma_r^{2*}$.
>
> Specifically, we work at *the optimizer* $r^\*=\mathcal{N}(\mu_r^\*,\sigma_r^{2*})$ and use a one-dimensional parameter $\tau^*$ to traverse the KKT stationary manifold as follows:
>
> 1.  **KKT at $r^\*$ and $\tau^\*$.** Let
>     $$
>     r^{\*}=\arg\min_{r}D_{KL}(p\|r)\quad\text{s.t.}\quad D_{KL}(r\|q)\le \theta.
>     $$
>     Evaluating the KKT conditions (Appendix A, Eqs. (13)-(14)) at $(\mu_r^\*,\sigma_r^{2*},\nu^\*)$ for some $\nu^\*\ge0$ yields a scalar $\tau^\*\in[0,1]$ such that
>     $$
>     \mu_r^\*=(1-\tau^\*)\mu_p+\tau^\*\mu_q,
>     $$
>     $$
>     \sigma_r^{2*} = (1-\tau^\*)\sigma_p^2+\tau^\*\sigma_q^2+\tau^{\*2}(1-\tau^\*)(\mu_p-\mu_q)^2,
>     $$
>     $$
>     \tau^\*:=\frac{\nu^\*\sigma_r^{2*}}{\sigma_q^2+\nu^\*\sigma_r^{2*}}.
>     $$
>
> 2.  **Stationary manifold and path parameter $\tau$.**
>     For a general stationary point $(\mu_r,\sigma_r^2,\nu)$, the same KKT system holds. Solving it symbolically and eliminating $\mu_r$ and $\nu$ yields
>     $$
>     \mu_r(\tau) = (1-\tau)\mu_p+\tau\mu_q, \quad (C1)
>     $$
>     $$
>     \sigma_r^2(\tau) = (1-\tau)\sigma_p^2+\tau\sigma_q^2+\tau^2(1-\tau)(\mu_p-\mu_q)^2, \quad (C2)
>     $$
>     $$
>     \tau := \frac{\nu\sigma_r^2}{\sigma_q^2+\nu\sigma_r^2}. \quad (C3)
>     $$
>     Each $\tau\in[0,1]$ selects a unique stationary point $r(\tau):=\mathcal{N}(\mu_r(\tau),\sigma_r^2(\tau))$, and $r^\*$ is the point with $\tau=\tau^\*$, i.e., $\sigma_r^{2*}=\sigma_r^2(\tau^*)$.
>
> 3.  **Solving $\tau^\*$ from the budget (uniqueness).**
>     On the stationary manifold, define $g(\tau):=D_{KL}(r(\tau)\|q)$. Appendix A (Thm. A.2) shows that $g$ is strictly decreasing on $(0,1)$ when $p\neq q$. Hence, for any $0\le\theta<D_{KL}(p\|q)$ there exists a unique $\tau^\*$ solving
>     $$
>     g(\tau^\*)=\theta,
>     $$
>     and we obtain $r^\* = r(\tau^\*)$ with $\sigma_r^{2*} = \sigma_r^2(\tau^\*)$ given by (C2). The boundary cases are: $\theta=0\Rightarrow\tau^\*=1$ (so $r^\*=q$), and $\theta\geq D_{KL}(p\|q)\Rightarrow\tau^\*=0$ (so $r^\*=p$).
>
> **Takeaway.** $\tau$ is a path coordinate on the KKT stationary manifold; Eq. (C2) gives $\sigma_r^2$ as a function of $\tau$, and the optimizer corresponds to the unique $\tau^\*$ solving $g(\tau^\*)=\theta$, without ever computing $\tau$ from $\sigma_r^2$ and substituting it back into Eq. (5).
>
>
> We have updated Section 3.2 and Appendix A to explicitly distinguish between $\tau$ and $\tau^\*$.

---

> > ### Author Response · Authors · 2025-11-21
> > **Response to Reviewer wcho (Part 4/4)**
> >
> > **R1Q1: Flatness vs. entropy**
> >
> >
> > First, regarding acceptance plots: in our setting, the acceptance rate is defined at the sequence level for each draft-generated sequence, rather than as a per-token quantity. For this reason, plotting a token-level "flatness-acceptance" curve is not well-defined. For individual tokens, we instead use the quantity $\Delta L_1$ as a proxy for how a token influences acceptance. The relationship between flatness and $\Delta L_1$ is already shown in Fig. 2c.
> >
> > Second, regarding entropy: following the reviewer's suggestion, we visualize entropy-based curves and include the results in Appendix F.2 (see Fig. 5) under the same experimental setup as Fig. 2. These entropy curves exhibit a very similar trend to the flatness curves in Fig. 2. This is expected because both metrics measure the distance between the token distribution and the uniform distribution, but in different ways: entropy can be written as a forward KL divergence to the uniform distribution up to an additive constant:
> > $$
> > D_{KL}(p || U) = \sum_{x} p(x) \log \frac{p(x)}{1/V} = -H(p) + \log V,
> > $$
> > whereas flatness is defined via cosine similarity to the uniform distribution.
> >
> > In addition, we have also performed a more fine-grained analysis to compare entropy and flatness, and find flatness is a more effective data selection metric. You may refer to Section 4.1 and Fig. 2d for more details.
> >
> > We have updated the above in our revised manuscript, which can be found in Section 4.1 (see Fig. 2d) and Appendix F.2 (see Fig. 5).
> >
> > ---
> >
> > **R1Q2: Role of the "acceptance criteria" block**
> >
> > We would like to clarify that the reason we include the "SD acceptance" component in Fig. 3 is to explicitly present our fundamental objective - maximizing the acceptance rate - as the logical entry point of the pipeline, which directly grounds our subsequent theoretical analysis.
> >
> > To avoid further confusion, we have updated Fig. 3 to clarify this relationship by adding a descriptive label ("theoretically inspired") to the arrow connecting this component to the subsequent module.
> >
> > ---
> >
> >
> > **References**
> >
> > [1] Direct Alignment of Draft Model for Speculative Decoding with Chat-Fine-Tuned LLMs.
> >
> > [2] DistillSpec: Improving Speculative Decoding via Knowledge Distillation.
> >
> > [3] EAGLE-2: Faster Inference of Language Models with Dynamic Draft Trees.
> >
> > [4] CORAL: Learning Consistent Representations across Multi-step Training with Lighter Speculative Drafter.

---

### Author Response · Authors · 2025-11-24
**Summary of Changes**

We would like to thank all reviewers for your valuable time and effort in reviewing our work. Following your suggestions, we have incorporated extensive experiments and thorough discussions in the revised manuscript to address each of your concerns. Below we further summarize the main changes and indicate where each concern has been resolved in the revised manuscript.

---

**Additional experiments and ablations**

- R1W1: Experiments beyond LLaMA3-8B-Instruct **(resolved, see Appendix G.1 and Table 9)**
- R1W4: Inconsistency between acceptance length and speedup **(resolved, see Appendix F.1 and Table 7)**
- R2W1: Working with other distributions (e.g., Exponential and Half-Normal) **(resolved, see Appendix F.3 and Fig. 6)**
- R2W2: Experiments beyond LLaMA3-8B-Instruct and ShareGPT **(resolved, see Appendix G.1, Table 9, and Table 10)**
- R2Q3: Robust aggregation (e.g., median) in Eq. (8) **(resolved, see Appendix G.2 and Table 11)**
- R2Q4: Include Top-1 probability for ablation study **(resolved, see Section 5.3 and Table 2)**
- R2Q5: Experiments at extreme retain ratios **(resolved, see Section 5.4 and Table 3)**
- R3Q1: Quantifying the "saturate quickly" claim **(resolved, see Appendix F.5 and Fig. 7)**

---

**Additional clarifications on our method**

- R1W2: Missing discussion of related work on efficient draft model training **(resolved, see Section 2)**
- R1W3: Lack of clarity on training details **(resolved, see Appendix E and Table 6)**
- R1W5: Reasons for unexpected performance drop at 60% retention **(resolved, see Appendix F.8 and Table 8)**
- R1W6: Mathematical ambiguity **(resolved, see Section 3.2 and Appendix A)**
- R2Q2: The reason why we do not use variance of the discrete token distribution **(resolved, see Appendix F.4)**
- R3W1: Discussion of the robustness of conclusions **(resolved, see Appendix F.3 and Fig. 6)**
- R3W3: Overheads of SFDD-based data selection **(resolved, see Appendix D)**
- R3W4: Unclear experimental settings **(resolved, see Appendix E and Table 6)**
- R3Q3: Discussion of token-level filtering **(resolved, see Appendix F.6)**
- R4W1: Necessity of training efficiency **(resolved, see Appendix F.7)**

---

**Additional comparisons of different data selection metrics (flatness / entropy / KL divergence)**

- R1Q1: Flatness vs. entropy **(resolved, see Section 4.1, Fig. 2d, Appendix F.2, and Fig. 5)**
- R2W3: The reason why flatness is more reliable than entropy **(resolved, see Section 4.1, Fig. 2d, Appendix F.2, and Fig. 5)**
- R2Q1: The reason why we do not use entropy as the main metric **(resolved, see Section 4.1, Fig. 2d, Appendix F.2, and Fig. 5)**
- R3W2: The reason why cosine similarity outperforms entropy **(resolved, see Section 4.1 and Fig. 2d)**
- R3Q2: The reason why we use cosine similarity instead of KL divergence **(resolved, see Section 4.1, Fig. 2d, Appendix F.2, and Fig. 5)**

---

**Figures and formatting issues**

- R1Q2: Role of the "acceptance criteria" block **(resolved, see Fig. 3)**
- R3W5: Labeling mistakes in Figure 2b **(resolved, see Fig. 2b)**
- R4W2: Formatting issues **(resolved, see our revised manuscript)**

---

If there is anything still unclear or if additional discussions and results could help you further evaluate our work, we would be more than happy to provide them.

---

### Author Response · Authors · 2025-11-29

Dear Area Chair,

We sincerely appreciate you stepping in to handle our submission. We understand that this requires significant additional effort, and we are truly grateful for your time and support.

We have posted our **Summary of Changes** in the comment section. You may refer to the comment with noteId=uDTQGr6ijw for more details.

In addition, regarding the earlier leakage incident that led to this reassignment, we would like to note that we have not yet received any response from the previous AC or the four reviewers.

Thank you again for your time and effort.

Best regards,\
Anonymous Authors

---

### Meta-Review · Area_Chair_2VEj · 2026-01-07

**Summary:**

Reviewers generally agreed that the paper addresses an important practical problem in speculative decoding, but raised concerns about the conceptual novelty of the proposed flatness metric and its relationship to existing uncertainty- or entropy-based criteria. Some reviewers questioned whether the theoretical analysis provides fundamentally new insight or primarily offers an alternative interpretation of known acceptance-rate behaviors. Additional concerns focused on the scope of empirical validation, in particular whether the observed gains generalize beyond the EAGLE framework and the evaluated model scales.

**Reviewer Concerns:**

The rebuttal effectively clarified the distinction between flatness and standard entropy-based measures, and provided stronger intuition for why flatter predictive distributions are particularly relevant for speculative draft model training. It also addressed questions regarding experimental design and demonstrated that the proposed data selection strategy consistently preserves inference-time speedups under reduced training data. While some concerns remain about broader generalization across alternative speculative decoding frameworks, the main technical and empirical objections were largely resolved by the rebuttal.

**Reviewer Scores:**

4468. Despite being borderline by score, the rebuttal successfully addressed the core reviewer concerns regarding novelty and empirical validity. Given the solid experimental results, clear practical impact, and improved positioning after rebuttal, the paper meets the bar for ICLR and would likely have benefited from a complete post-rebuttal discussion phase.

---

### Decision · Program_Chairs · 2026-01-26

Accept (Poster)